# Cell cycle criticality as a mechanism for robust cell population control

Benjamin D Simons [1,2,3] & Omer Karin [4✉]

## Abstract

**Tissue homeostasis requires a precise balance between stem cell self-renewal and differentiation. While fate decisions are known to be closely linked with cell cycle progression, the functional significance of this relationship is unclear. We propose a mechanistic framework to analyse cellular dynamics when cell fate is coupled to cell cycle duration. Our model highlights a unique aspect of cell cycle regulation where mitogens serve as control parameters for a bifurcation governing the G1-S transition. Under competitive feedback from cell–cell interactions, the cell cycle regulatory network fine-tunes near the critical point of this bifurcation. Critical positioning lengthens G1 while amplifying cell-to-cell variability in mitogenic signalling and biochemical states. Such regulation confers significant advantages for controlling cell population dynamics, with alternative topologies enabling rapid tissue growth and repair or efficient mutant rejection. Counter-intuitively, we propose that stem cells may couple prolonged G1 with increased self-renewal propensity to efficiently suppress mis-sensing mutants. Our theory provides a distinct explanation to dynamical and statistical patterns of G1 lengthening and predicts regulatory strategies across development, homeostasis, and ageing.**

**Keywords** Cell Cycle; Stem Cells; Tissue Homeostasis; Mathematical Modelling; Bifurcation
**Subject Categories** Cell Cycle; Computational Biology

## Introduction

Tissues are formed and maintained through a dynamic interplay of cell division, differentiation, and loss. Achieving the correct tissue size and composition requires a precise balance of these processes. Many tissues, such as blood and renewing epithelia, are sustained by multipotent stem and progenitor cells (Beumer and Clevers, 2024; Kitadate et al, 2019; Orkin and Zon, 2008). In such tissues, cell loss or changes in the demand for specific cell types trigger compensatory processes of cell division and differentiation within

the stem and progenitor populations, thereby restoring tissue homeostasis. Similarly, in developing tissues, progenitor populations must modulate their division and differentiation rates as the tissue grows to attain the appropriate size and composition. The problem of cell population control is also crucial for engineering synthetic circuits with applications in industry and biomedicine (Balagaddé et al, 2005; Glass et al, 2024; Ma et al, 2022; You et al, 2004).

Functionally, cell fate control circuits must rapidly and precisely correct deviations from set points while suppressing the expansion of mutants that mis-sense signals related to cell division or removal. Experimental and theoretical work has demonstrated that cell–cell negative feedback plays a central role in this regulation (Fan and Meyer, 2021; Karin and Alon, 2017; Kitadate et al, 2019). In this mechanism, population composition modulates the local microenvironment through biochemical or mechanical feedback, thereby influencing subsequent fate decisions through the modulation of intracellular regulation mechanisms. However, the design principles of the intracellular mechanisms that ensure the robust and efficient operation of these population control circuits remain unclear.

In this study, we propose that cells employ a *temporal* mechanism for cell fate regulation, where stem cells exploit a coupling between the cell division rate and cell fate choice to implement robust cell population control. While the processes regulating these rates may, in principle, be independent, in many systems they appear to be tightly coupled. The regulation of the cell division rate occurs primarily through controlling the average length of the G1 phase until transition into the S phase, in a process known as cell cycle entry. Major signalling pathways that control cell division rate through adjustment of the duration of the G1 phase of the cell cycle (mitogens), such as Wnt/$\beta$-catenin and MAPK/ERK, are also key regulators of cell identity (Blagosklonny and Pardee, 2002; Clevers et al, 2014; Crespo and Leon, 2000; Filmus et al, 1994; Lavoie et al, 2020; Niehrs and Acebron, 2012). Increasing experimental evidence suggests that the length of G1 itself controls cell identity and differentiation (Boward et al, 2016; Dalton, 2015; Liu et al, 2019; Zhao et al, 2020). This regulatory mechanism has been demonstrated in various contexts including embryonic stem cells, where G1 length determines fate choice (Becker et al, 2006; Calder et al, 2013; Coronado et al, 2013; Jang et al, 2019; Liu et al, 2019; Pauklin and Vallier, 2013; Perera et al, 2022); during neurogenesis (Artegiani et al, 2011; Hardwick et al,

[1]Department of Applied Mathematics and Theoretical Physics, Centre for Mathematical Sciences, University of Cambridge, Cambridge CB3 0WA, UK. [2]Wellcome Trust, Cancer Research UK Gurdon Institute, University of Cambridge, Cambridge CB2 1QN, UK. [3]Wellcome Trust-Medical Research Council Cambridge Stem Cell Institute, Jeffrey Cheah Biomedical Centre, University of Cambridge, Cambridge CB2 0AW, UK. [4]Department of Mathematics, Imperial College London, London SW7 2AZ, UK.
✉E-mail: o.karin@imperial.ac.uk

2015; Hindley and Philpott, 2012; Lange et al, 2009; Lim and Kaldis, 2012; Molina and Pituello, 2017; Pilaz et al, 2009); in adult stem cell populations (Carroll et al, 2018; Freije et al, 2012; Gandarillas, 2012; Johnson et al, 2020; Mende et al, 2015; Treichel and Filippi, 2023; Wilson et al, 2004); and in terminal differentiation (Latella et al, 2001; Ruijtenberg and van den Heuvel, 2016; Zhao et al, 2020). Despite its apparent fundamental importance in modulating cell fate choice in tissues, the implications of the coupling between G1 lengthening and cell fate choices as a mechanism for cell population control remain unclear.

Here, we develop a mechanistic framework to analyse this coupling. We demonstrate that G1 lengthening occurs as a noisy saddle-node bifurcation, where regulatory signals tune a bifurcation parameter to modulate G1 duration. We show that when G1 length is coupled to cell fate by negative feedback, the dynamics leads generically to self-tuning to the vicinity of a critical bifurcation point. This self-tuning results in convergence to a robust fixed point of the population size. The critical mechanism can provide the cellular population with rapid responses to deviations from this fixed point or with efficient rejection of mis-sensing mutants, explaining experimental observations on cell cycle dynamics. We demonstrate that these two features provide a trade-off with each other and are favoured by alternative topologies, explaining patterns of regulation in development, homeostasis, and ageing.

## Results

### G1 lengthening occurs by critical dynamics that amplifies variation between cells

The G1-S transition is governed by a molecular regulatory network that involves interactions between the proteins Rb, E2F, CycD-CDK4/CDK6 (abbreviated CycD), and CycE-CDK2 (abbreviated CycE) (Barr et al, 2016; Blagosklonny and Pardee, 2002; Novák and Tyson, 2004, 2021; Tyson and Novak, 2014; Yao et al, 2008) (Fig. 1A). The transition occurs following an increase in the activity of E2F and the transcription of CycE proteins. The underlying dynamics correspond to a bistable activity pattern due to positive feedback (Cappell et al, 2016; Yao et al, 2008). Active Rb protein forms a complex with E2F that represses its activity and prevents CycE transcription, while CycE in turn, reduces the activity of Rb by phosphorylation, closing the positive feedback loop. The feedback between these proteins can stabilize both a low E2F state, and a high E2F state that is associated with the G1-S transition.

The transition from the low E2F state to the high E2F state is also controlled by the activity of another protein, CycD (Novák and Tyson, 2004; Schwarz et al, 2018; Tyson and Novak, 2014; Yao et al, 2008). CycD affects the transition of E2F from off to on by also reducing the activity of Rb via phosphorylation (Topacio et al, 2019). When CycD is high, a bifurcation occurs that destabilizes the low E2F state and transitions the system to the high E2F state, promoting the G1-S transition. CycD levels are nearly constant throughout G1 and are controlled by a wide range of external signals, including through the MAPK and Wnt/$\beta$-catenin pathways (Hitomi and Stacey, 1999; Zerjatke et al, 2017). Thus, a low level of CycD extends G1 as it prevents the G1-S transition, while a high level shortens G1.

While the bistable nature of the Rb/E2F switch is well-established (Cappell et al, 2016; Konagaya et al, 2024; Weinberg, 1995; Yao et al, 2008), there is debate regarding the precise biological mechanisms underlying its implementation (Konagaya et al, 2024; Narasimha et al, 2014; Yao et al, 2011; Zerjatke et al, 2017). We therefore chose to focus on capturing the essential dynamics through a minimal framework, noting that our subsequent analysis applies to a much broader class of possible models. Denoting the level of active (unphosphorylated) Rb protein as $r$, its dynamics are governed by a constant production rate and removal through both degradation, at rate $\gamma_D$, and phosphorylation, at rate $\gamma_P$. The phosphorylation rate depends on both CycD activity $C$ and a feedback loop driven by the level of $r$ itself, which we capture phenomenologically by the relation $\gamma_P(r, C) = V \frac{C}{1+C/k_1} \frac{1}{1+(r/k_2)^2}$, where $k_1$ controls the scale at which CycD activity becomes saturated, $k_2$ fixes the level at which Rb modulates its phosphorylation rate, and $V$ fixes the net amplitude. We assume that the feedback of $r$ on itself, which occurs through the E2F/CycE cascade, is cooperative. We also incorporate the impact of noise, such as that arising from stochastic fluctuations in the levels of different factors that control Rb phosphorylation. These fluctuations can be modelled as a Wiener-type process with, for example, a multiplicative noise amplitude $\sigma r$ (it is important to notice that the precise form of the noise will be inconsequential for our conclusions). The dynamics of $r$ are therefore given by the stochastic differential equation,

$$dr = \left(1 - (\gamma_D + \gamma_P(r, C))r\right) dt + \sigma r \, dW, \qquad (1)$$

where $dW$ denotes a Wiener process (Fig. 1B).

Within this framework, the essential dynamics of the circuit are captured by a saddle-node bifurcation, with the activity level of CycD, $C$, serving as a control parameter for the bifurcation. The bifurcation, which is depicted graphically in Fig. 1B, is the key component of a wide range of mathematical models for the G1-S transition, from simple one-dimensional models to highly elaborate mechanistic models (Barr et al, 2016; Kwon et al, 2017; Stallaert et al, 2019; Yao et al, 2008) (Methods). The control parameter $C$ (affected by mitogenic signals) destabilizes the state associated with G1, resulting in the G1-S transition. When levels of $C$ are low, the system is positioned in a stable configuration (G1) and is resistant to fluctuations. At higher levels of $C$, this phase becomes metastable with the emergence of a second fixed point of the dynamics. Eventually, at a critical level of CycD activity $C = C_{\text{crit}}$, the system transitions rapidly into S phase. Importantly, we assume that $C$ is generally constant during G1, and later consider the case where $C$ changes in a time-dependent manner.

The saddle-node bifurcation has three key properties that will play an important role in our analysis. The first is that in the vicinity of $C \approx C_{\text{crit}}$, the dynamics of the system are captured by a one-dimensional noisy process,

$$dx = (x^2 + \mu) dt + \xi \, dW, \qquad (2)$$

where $x$ denotes the rescaled coordinates that parametrize the bistable switch, $\xi$ corresponds to noise amplitude by the critical point, time $t$ is measured in rescaled coordinates, and $\mu \propto C - C_{\text{crit}}$ is

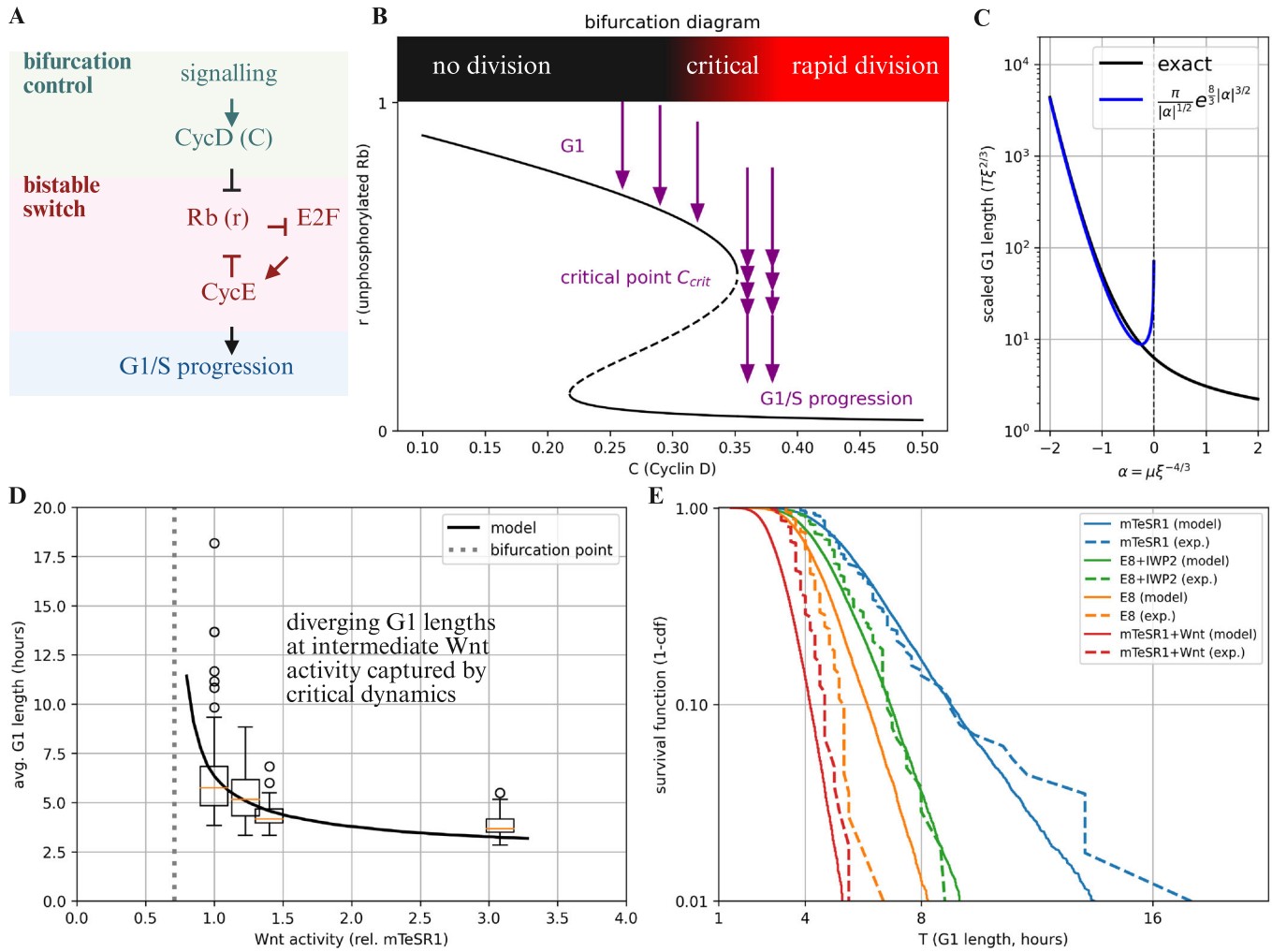

**Figure 1.  Model for G1 lengthening.**

(A) Outline of the circuit that controls G1-S progression during cell cycle, with signalling input controlling a positive-feedback-based mechanism. (B) Bifurcation diagram for the G1-S progression mechanism showing how changes in CycD concentration mediate an abrupt change of unphosphorylated Rb concentration (see main text). (C) Dependence of the G1 length $T$ on the scaled distance from the bifurcation $\mu$. The black line shows the exact solution for the mean delay (see Methods). The blue line shows the exponential approximation (Kramers' formula) valid for $\mu < 0$. (D) G1 length distribution as a function of intracellular Wnt activity (as reported by AXIN2 versus GAPDH expression, relative to lowest reported activity level) for hESCs grown in the following experimental conditions: mTeSR1 media ($n = 114$), E8 media supplemented with IWP2 ($n = 105$; IWP2 is a Wnt inhibitor), mTeSR1 media supplemented with Wnt ($n = 104$), and E8 media ($n = 112$). The G1 length shows a sharp and highly nonlinear increase associated with increased variability around a specific Wnt activity level (data from ref. Jang et al (2019)). This sharp increase is consistent with a G1 elongation mechanism based on a noisy saddle-node bifurcation. The black line corresponds to (average) model simulations with the bifurcation control parameter proportional to Wnt activity, taking a critical concentration at 3/4 of the minimal observed Wnt activity level (dashed grey line) with fixed cell-to-cell variability in the control parameter (Methods). For the box plots, the box extends from the first quartile to the third quartile, with a line at the median. The whiskers extend to the farthest data point within 1.5 times the inter-quartile range from the box. Outliers beyond the whiskers are plotted as individual points (fliers). (E) Comparison of variability in G1 lengths between data from ref. Jang et al (2019) and model simulations. The model captures the sharp increase in variability associated with G1 lengthening, which is due to critical dynamics amplifying cell-to-cell variation. See the Methods for simulation details and parameters.

the (rescaled) distance from the bifurcation. The G1 state is associated with negative values of $x$, while the G1-S transition corresponds to the irreversible progression of $x$ to positive values. The length of G1, denoted $T$, is the time taken for this transition, which can be prolonged by distinct mechanisms depending on the sign of $\mu$. For $\mu < 0$ (sub-critical), the system resides in a stable G1 state, and the transition is a stochastic event requiring a noise-driven escape over an unstable barrier; here, $T$ corresponds to the mean first passage time. Conversely, for $\mu \geq 0$ (super-critical), the G1 state is no longer stable, but, as long as $\mu$ is small, the system's

deterministic progression may be significantly delayed as it passes through a "ghost" of the former fixed points, a phenomenon known as critical slowing down (Koch et al, 2024; Strogatz, 2024). Otherwise, for large $\mu$, G1 progression is rapid.

The second property is the strong sensitivity of the length of G1, $T$, around the critical point, which is marked by an abrupt, super-exponential increase at the critical threshold (with $\log T \sim |\mu|^{3/2}$) (Karin et al, 2023; Simons and Karin, 2024) (Fig. 1C). As such, any prolonged G1 length is predicted to occur when the regulatory network governing the G1-S transition is poised in the vicinity of

the critical point $C \approx C_{crit}$, with $\mu \approx 0$, where the dynamics are captured by Eq. (2).

The third property relates to the variation between individual cells and between signalling and media conditions. Different media conditions are associated with different levels of mitogenic signalling. Cellular heterogeneity introduces additional variation arising from differences in cellular physiology, biochemical composition, and spatial configuration. These variations are captured through variation in the coordinate $\mu$. Crucially, the steep lengthening of $T$ and its strong sensitivity to $\mu$ around the critical point (Fig. 1C) amplifies cell–cell variation and variation between conditions, resulting in steep and divergent G1 lengthening around $\mu \approx 0$.

To test whether experimental data is consistent with G1 lengthening occurring through a noisy saddle-node bifurcation, we analysed published data on the distribution of G1 duration and its regulation in human embryonic stem cells (hESCs) (Jang et al, 2019) (Fig. 1D,E). G1 phase length in hESCs depends on culture media conditions, including signalling factor concentrations, such as Wnt signalling through the Wnt/$\beta$-catenin pathway (Becker et al, 2006; Calder et al, 2013; Coronado et al, 2013; Jang et al, 2019; Liu et al, 2019; Pauklin and Vallier, 2013).

In our model, activation of the Wnt pathway modulates the control parameter $C$ (Fig. 1D). As mitogenic signalling approaches a critical threshold, the model predicts a nonlinear rise in mean G1 duration, accompanied by a sharp increase in cell-to-cell variability. Experimental results from Jang et al on hESCs under varying Wnt conditions confirm these predictions: both the mean G1 length and its variance diverge within a narrow range of Wnt activity, in line with model predictions (Fig. 1E). The bifurcation-based model captures the experimental data for mean G1 length ($R^2 = 0.86$) and variance ($R^2 = 0.92$), substantially outperforming linear regressions ($R^2 = 0.62$ and $R^2 = 0.34$, respectively; Fig. EV1).

The model captures the striking divergence in G1 lengths that accompanies the modest increase in mean G1 duration at lower Wnt signalling levels (Fig. 1E). In E8 medium, hESCs exhibit a tightly clustered G1 length distribution with an average of $T \approx 4.3$ h. In mTeSR1 medium, the mean G1 duration increases to $T \approx 6.3$ h— a ~50% increase—yet variability rises far more dramatically, with the coefficient of variation increasing by ~250% due to a heavy tail of cells with very long G1 phases (>6 h). Similar variability amplification occurs when G1 length is perturbed by Wnt agonists or inhibitors. The model captures this behaviour as a consequence of critical dynamics near a noisy saddle-node bifurcation. Together, these findings support a framework in which G1 lengthening arises from bifurcation-driven amplification of variability.

A key prediction of our model is that, when poised in the vicinity of the critical point ($C \approx C_{crit}$), the system can enter a prolonged and fluctuating intermediate state before committing to S-phase. This behaviour is due to critical slowing down of the dynamics (Fig. 2A), a phenomenon also known as a "ghost" transient (Karin et al, 2023; Koch et al, 2024; Strogatz, 2024). These transients have a distinct phenomenology: G1 progression becomes multi-phasic, with a long and highly variable initial phase at an intermediate Rb-E2F activity level, followed by a rapid and more uniform transition into S-phase. Our model predicts that the duration of this intermediate phase is highly sensitive to the precise level of the mitogenic signal, $C$, and that this state is reversible; cells lingering here have not yet passed an irreversible commitment point and will revert to quiescence upon receiving inhibitory signals.

This predicted phenomenology is in excellent agreement with recent experimental findings on the Rb-E2F activity state in slowly dividing cells (Konagaya et al, 2024). Our model recapitulates the key features observed experimentally, including a prolonged and noisy plateau at an intermediate activity level (Fig. 2B, compare with Fig. 2C in Konagaya et al (2024)) and a uniform exit from this state to the S transition (Fig. 2C, compare with Fig. 2E in Konagaya et al (2024)). Moreover, our simulation of an inhibitory perturbation correctly predicts that cells in the intermediate state can either revert to quiescence or proceed to S-phase, depending on their position along the trajectory at the moment of inhibition (Fig. 2D, compare with Fig. 2K in Konagaya et al (2024)). Collectively, these results suggest that ghost state dynamics provide a robust mechanistic explanation for the experimentally observed intermediate state that safeguards proliferation commitment.

## Tuning of G1 length by competition provides a set-point for self-renewing tissues

The observation that cell fate decisions may require specific lengthening of G1 raises questions about how this lengthening is achieved in tissues. We focus here on a scenario in which a cell, such as a tissue stem cell, must select between self-renewal and differentiation through symmetric cell divisions, with this decision depending on the duration of G1 length $T$. (Note that the model can readily generalize to other patterns of cell fate, such as choosing between alternative fates in developmental settings.) The strong sensitivity of G1 length on the bifurcation parameter $C$, and consequently to external signals such as Wnt signalling, suggests that for an arbitrary parametrization, G1 length will be either very short or effectively infinite, implying a strong imbalance between self-renewal and differentiation. Denoting by $N$ the number of undifferentiated cells in the population, we can capture the net growth of $N$ by:

$$\frac{\dot{N}}{N} = T^{-1}\theta(T), \tag{3}$$

This equation describes the rate of change for the stem cell population $N$ at a fixed cell division rate. The parameter $T$ corresponds to the G1 duration of a cell, which is assumed to dominate the overall cell division time, so the term proportional to $1/T$ converts the outcome of a discrete cell division event into a continuous growth rate. The function $\theta(T)$ captures the net self-renewal bias, which is defined by the probabilities of the different fate outcomes. Denoting the probability of a self-renewing division as $P_R(T)$ and a differentiating division as $P_D(T)$, the bias is given by $\theta(T) = P_R(T) - P_D(T)$. For instance, if a cell division of duration $T$ yields on average the same number of differentiated and stem cells, then $\theta(T) = 0$. Critically, our model does not presuppose a specific relationship between G1 length and cell fate. Instead, as we will show, the nature of this dependence is what defines two distinct and functionally opposing regulatory strategies.

To see how G1 lengthening can occur, we note that cells typically share and modulate environmental signals, including the concentrations of the molecules that adjust G1. This results in competition and negative feedback, where a change in the abundance or composition of the cell population by G1 lengthening or shortening feeds back on the environmental signal. As an

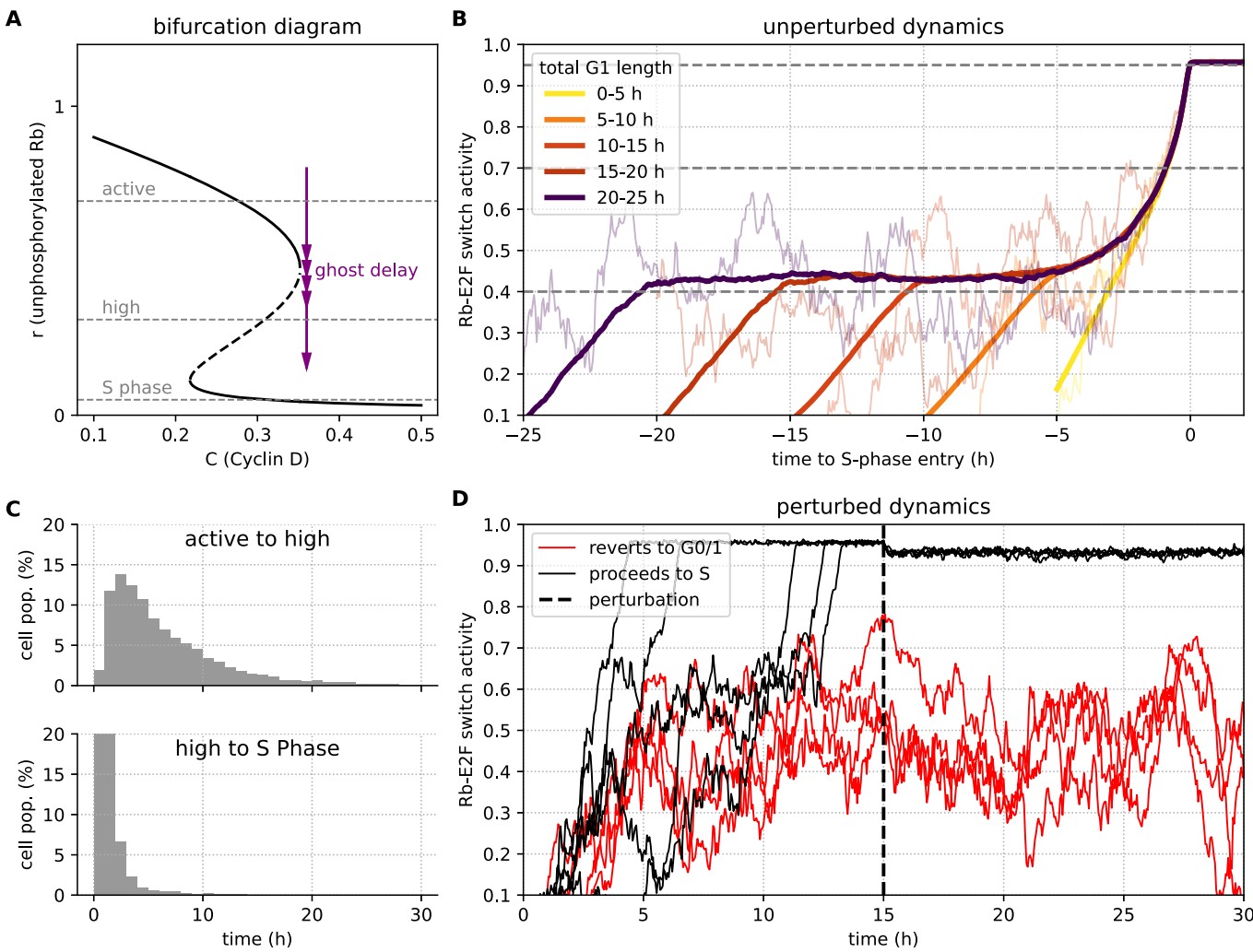

**Figure 2.    Model simulation of the intermediate Rb-E2F state and its reversibility.**

(A) Bifurcation diagram of the model. The purple arrows illustrate the trajectory of a cell near the critical point, where it experiences a significant "ghost delay" as it passes the remnant of the stable and unstable fixed points. We mark on the diagram three thresholds (dashed grey lines) that characterise the dynamics—an active activity threshold marking the transition entering the ghost region (corr. to an active state of the Rb/E2F network in Konagaya et al (2024)), a high Rb/E2F activity threshold marking the exit from the ghost region (corr. to CDK2 activity of 0.65 in Konagaya et al (2024)) and an activity threshold associated with S phase entry. (B) Average trajectories of the Rb-E2F switch activity (proxied by $1 - \tanh(r)$) for cells binned by their total G1 length (coloured lines), aligned to the moment of S-phase entry ($t = 0$). Lighter, thin lines show a single sample trajectory from each bin to illustrate cell-to-cell variability. (C) Histograms showing the distribution of time spent in two qualitatively distinct phases of G1 for a population of simulated cells. The top panel shows the distribution of durations from the initial crossing of the active threshold to the initial crossing of the high activity threshold. The bottom panel shows the subsequent, more rapid transition from the high activity threshold to S phase threshold. (D) Simulation of an inhibitory perturbation. Cells progressing through G1 are subjected to a sudden drop in $C$ at $t = 15$ h (dashed line). Trajectories show that cells that have not yet passed a commitment point revert to a quiescent state (red), while committed cells proceed to S phase (black), demonstrating the reversibility of the active-intermediate state. See the Methods section for simulation details and parameters.

example, consider the case where the cell population size or density $N$ modulates the control parameter for the bifurcation,

$$C = f(N), \qquad (4)$$

where $f$ may correspond to the effect of competition over secreted factors and may be affected by contact inhibition. For simplicity, here we assume that differentiated cells exit the population (though this assumption is not essential, as the dynamical patterns we discuss also emerge in more complex regulatory circuits; in the Methods section we provide a specific

mechanistic example where this equation arises, and analyse an alternative case where feedback arises from the differentiated cell population (Buzi et al, 2015; Lander et al, 2009; Lo et al, 2009; Manceau et al, 2008; Tata and Rajagopal, 2016)). In this case, a balanced state can occur when $N$ is constant, $\dot{N} = 0$. Here, the G1 duration $T = T_S$, where the growth rate of $N$ is zero, $\theta(T_S) = 0$. Since every finite and extended $T_S$ occurs at the vicinity of $C \approx C_{\mathrm{crit}}$, where the cell cycle length is prolonged, the set-point of the self-renewing population becomes

$$N_S \approx f^{-1}(C_{\mathrm{crit}}). \qquad (5)$$

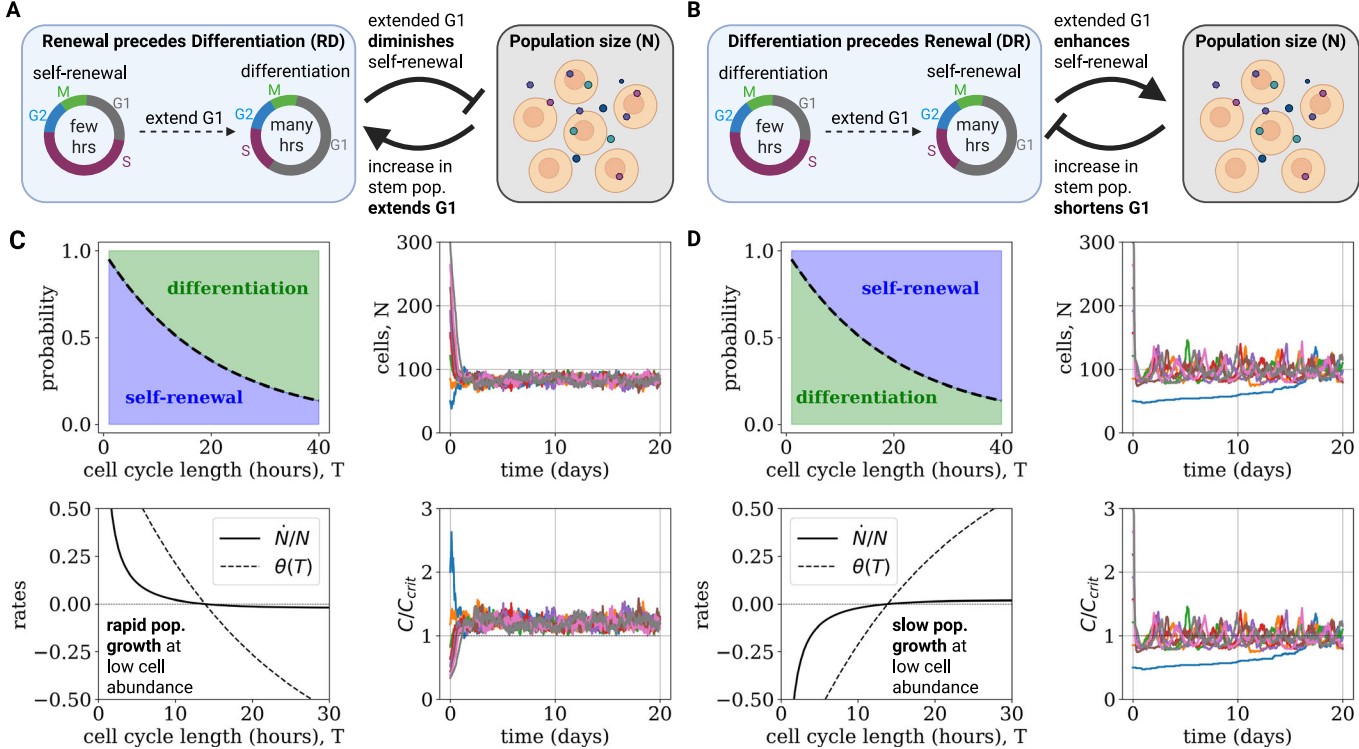

**Figure 3. Set point regulation and alternative topologies.**

(A) RD topology, with rapid cell division favouring self renewal. (B) DR topology, with rapid cell division favouring differentiation. (C) Simulations of population dynamics under the RD topology from different initial concentrations, with feedback set by $C = f(N) = C_{crit}N_{st}/N$ with $N_{st} = 100$, demonstrating convergence to the critical set point (right subpanels). In the left subpanels, we plotted the probability that a cell division will result in duplication (self-renewal) versus differentiation as a function of cell cycle length ($T$) and the effective rate functions $\theta(T)$ and $\dot{N}/N = T^{-1}\theta(T)$ (see main text). To account for additional processes that can drive cell division, such as injuries or physiological changes, all simulations included, in addition to regulated cell divisions, random fixed-rate cell divisions. (D) Corresponding circuit and simulations for the DR topology (here $C = f(N) = C_{crit}N/N_{st}$). Note the slow recovery at low initial population, which is driven by random cell divisions. In these simulations we chose a steady-state stem cell niche of $N = 100$; similar dynamics occur even when the stem cell niche is very small and includes only $N = 10$ cells as we demonstrate in Fig. EV2. See the Methods section for simulation details and parameters.

This fixed point is stable if around the fixed point we have that

$$\text{sign}\left[\frac{\partial\theta}{\partial T}\right] = \text{sign}\left[\frac{\partial f}{\partial N}\right]. \qquad (6)$$

This equality holds for two distinct topologies. In the first case, an increase in cell abundance prolongs G1, requiring that self-renewal preferentially occurs at shorter G1 lengths. This topology will henceforth be referred to as the Renewal precedes Differentiation (RD) topology (Fig. 3A). The RD topology could be implemented in cells, for example, through a slow G1-dependent accumulation of a differentiation factor, as occurs in adipocyte differentiation (Zhao et al, 2020), or through G1-dependent susceptibility to receiving differentiation signals. Alternatively, an increase in $N$ may preferentially *shorten* the length of G1, coupled with increased differentiation at shorter G1 lengths. This can occur if cells need sufficient time in G1 to maintain their identity, such that early entry into the cell cycle increases the likelihood of differentiation. For example, if a differentiating factor is degraded in G1 and accumulates in other cell cycle phases, or through G1-dependent susceptibility to signals required for maintenance of cell identity. We denote the second case as the Differentiation precedes Renewal (DR) topology (Fig. 3B).

While both the RD and DR topologies exhibit sensitive responses to changes in cell abundance, they have very different dynamics away from the set point. When the number of cells $N$ is much lower than the set point, both topologies can result in pure self-renewal, but the dynamics differ. In the RD topology, self-renewal occurs rapidly due to short cell cycle times. Only as the population approaches its steady state does a slowdown in G1 occur, resulting in large-scale cell cycle arrest (Fig. 3C). In control theory, this is known as *bang-bang control*, providing an optimal response for reaching a set point in minimal time (Itzkovitz et al, 2012). This type of dynamics appears to be widespread in developmental settings (Itzkovitz et al, 2012; Kicheva et al, 2014; Lange et al, 2009; Molina et al, 2022). Conversely, the DR topology exhibits sluggish recovery from low cell numbers due to the extension of division times (Fig. 3D). However, when the population is too large, the DR topology can respond rapidly with short cell cycles that promote differentiation. While the RD topology might seem disadvantaged in this scenario due to its longer cell cycles, this can be easily circumvented if cells commit to differentiation before G1 is concluded, as is common in terminal differentiation. We therefore propose that the RD topology provides a particularly effective mechanism for rapid tissue development and growth.

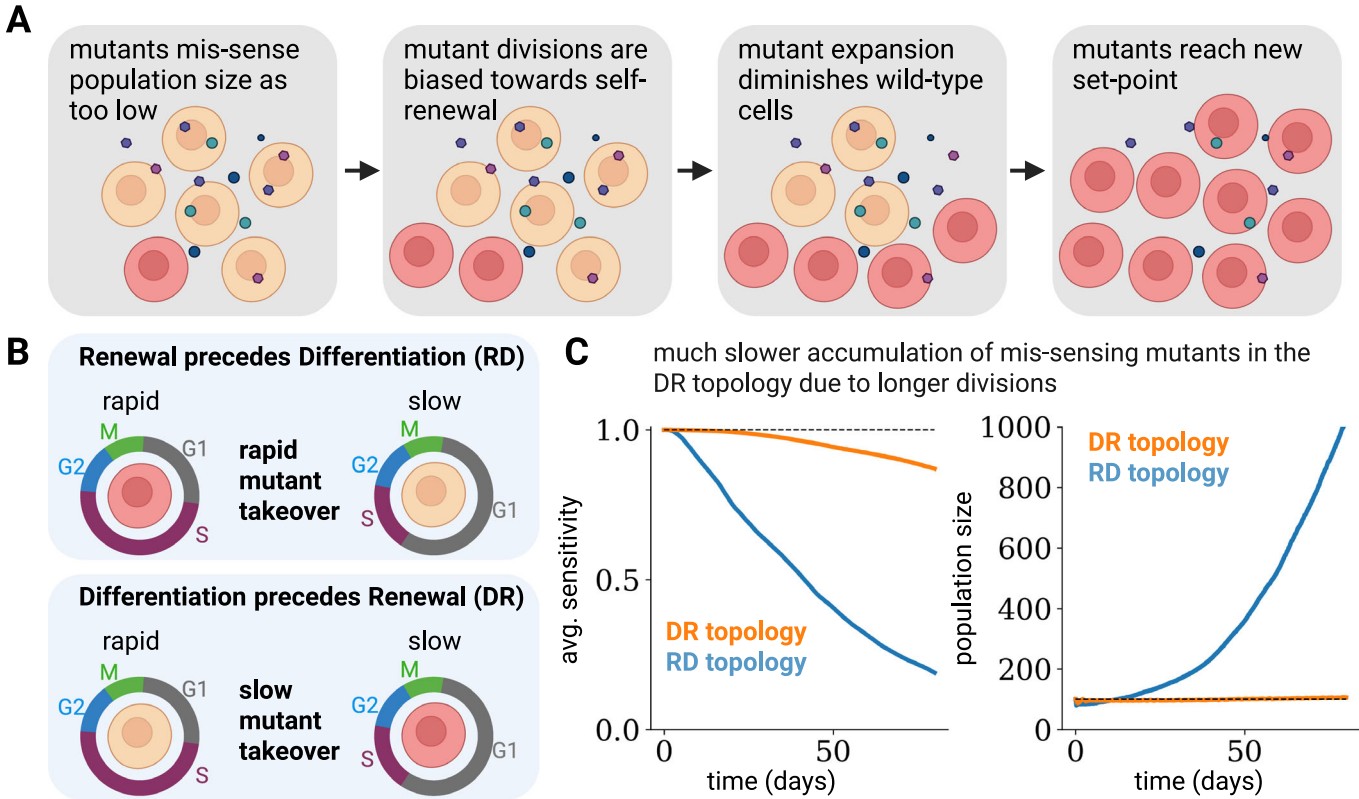

**Figure 4. Efficient mutant rejection in the DR topology.**

(A, B) RD and DR dynamics show differential dynamics, with the sharp sensitivity around the critical point ensuring that mis-sensing mutants will either have rapid or prolonged cycling times, respectively. (C) Population dynamics were simulated with mutations as follows. $f(N)$ was replaced in the simulations by $f(a_i N)$, with $a_i$ denoting the sensitivity of cell $i$, initialized at unity. At each cell division, $a_i$ was "mutated" with probability $p_m$ or inherited with probability $1 - p_m$, with mutation corresponding to multiplication of $a_i$ by a random number drawn from a distribution $\mathcal{F}$. (Here, $\mathcal{F}$ was taken as a log-normal distribution) Both the RD and DR topologies drift towards mutants that under-sense the population size and thus have a higher population fixed point. However, the DR topology is more resistant, as its invading mutants are associated with longer cell cycle durations. See the Methods section for simulation details and parameters.

## Mutant rejection versus growth regulation determines design principles of tissue circuits

Within a steady-state environment, the fate decisions of wild-type cells are such that cell divisions are equally likely to result in differentiation or self-renewal. In such an environment, mutant cells that mis-sense input signals in such a way that they perceive the population size as too low would be biased towards self-renewal. This bias leads to their expansion within the population, ultimately displacing the wild-type cells and shifting the system's set-point (Fig. 4A). Preventing such mutant takeover is therefore a fundamental challenge for population-control circuits.

Formally, within the modelling framework, mutants can have altered sensing or processing of growth signals, leading to an altered mapping of cellular composition to growth regulation, captured by a dependence of the control parameter on $N$,

$$C = g(N). \tag{7}$$

This results in a different population set-point for the mutants $N'_s = g^{-1}(C_{\text{crit}})$.

The strong sensitivity of cell fate around the critical point provides an additional benefit: the efficient rejection of "cheater" mutants that have aberrant sensing or processing of regulating signals. To see how the sensitivity in the critical region contributes to mutant rejection, consider the case where an individual mutant is induced on the steady-state background of wild-type cells $N = N_s$. These mutants will have either a very short G1 ($T$ on the order of a few hours) or a very long G1 length. Efficient mutant rejection only requires that the growth rate $T^{-1}\theta(T)$ will be low at short G1 and long G1 lengths. As we will see, this requirement manifests differently for the DR and RD topologies.

The condition for efficient mutant rejection is achieved naturally in the DR topology (Fig. 4B,C). Here, mutants with shorter G1 length are lost due to differentiation. Mutants with an extended G1 phase may accumulate, but this process occurs over a long timescale due to their much slower division rate. This is a direct consequence of operating near the critical point, where a small mutational change in signal sensitivity is amplified into a very large increase in G1 duration (Fig. 1C), rendering the mutants less competitive. Consequently, the model predicts a gradual accumulation of clones with impaired responsiveness to signals that typically lengthen the G1 phase duration for this regulatory topology.

In contrast, the RD topology is fragile to the invasion of mutants (Fig. 4B,C). Mutants that are associated with a shorter G1 phase have both a shorter cell cycle (and thus, most likely, a faster overall division rate) and a higher self-renewal potential. As these mutants expand in cell number, they may alter their environment in such a way that they confer a growth disadvantage on their neighbouring wild-type cells, eventually resulting in the elimination of the wild-type population.

There are several ways that evolved systems may mitigate the detrimental aspects of mutant clone invasion associated with the RD topology. One possibility is that there is an overall compensation of S, G2, and M lengths to shorter G1, which indeed appears to be the case for embryonic stem cells (Jang et al, 2019). Another possibility is to engineer biphasic regulation of the growth by G1 length, where very short G1 lengths can result in cell removal (Fig. 4C), due, for example, to insufficient time to prepare for DNA replication (Ahuja et al, 2016). In this case, mutants with very short G1 will be eliminated. Biphasic regulation leads to the emergence of an additional unstable fixed point at short G1 lengths, and can thus result in a runaway elimination of the cell population following large perturbations (Karin and Alon, 2017), which can be detrimental for tissue development or regeneration.

The advantages of the DR topology for mutant rejection suggest that it may offer significant benefits for regulating self-renewal in tissues requiring long-term replenishment. It is well-established that in many such tissues, including the human blood, skin, and gut, self-renewal depends on a slow-cycling stem cell population with long-term self-renewal potential. This population transitions into a more rapid-cycling transit-amplifying (TA) population with limited self-renewal potential (Rangel-Huerta and Maldonado, 2017). For the DR topology to play a role in the dynamics of these systems, one would expect that a shortening of the G1 length would *drive* entry from the stem to the TA compartment. While not conclusive, evidence suggests that, across several self-renewing tissues, genetic changes that increase proliferation serve to diminish the size of the stem cell pool, whereas inhibition of proliferation enhances self-renewal at the expense of cellular differentiation (Freije et al, 2012; Gandarillas, 2012; Pietras et al, 2011; Watt et al, 2008; Wilson et al, 2004; Yamada et al, 2013). Similarly, during the ageing of self-renewing tissues, there appears to be an accumulation of cells with improved self-renewal coupled to diminished cell division rate (Abby et al, 2023; Hammond et al, 2023; Martincorena et al, 2018).

The DR topology is predicted to be associated with the accumulation of slow-cycling mutants with high capacity for self-renewal (Fig. 4C). One system where clonal dynamics have been characterized during ageing is the blood, where a few clones with specific mutations accumulate during ageing, in a phenomenon known as clonal haematopoiesis (Jaiswal and Ebert, 2019). The model specifically predicts that this process will be driven by cells that have prolonged cell cycle duration and that are associated with a higher set point $N_s' > N_s$. Both predictions appear to hold for the haematopoietic system as, with age, haematopoietic stem cells appear to grow in number (Geiger et al, 2013; Rossi et al, 2005), have longer cell cycle times driven by a longer G1 phase (Hammond et al, 2023), and are less sensitive to mitogens (Hammond et al, 2023). This suggests the possibility that ageing of haematopoietic stem cells is driven by competition and the accumulation of slowly dividing cells with impaired sensitivity, which are hallmarks of the DR topology.

The RD topology, on the other hand, appears to be widespread in developmental contexts. Here, we expect tissues to be fragile to the invasion of rapidly dividing and self-renewing mutant cells. This phenomenon has been experimentally demonstrated, for example, after CycD + CDK4 overexpression in neurogenic progenitors (Lange et al, 2009), and may also be relevant for various mutants in the context of cell competition during development (Baker, 2020).

The model thus predicts that, while the RD topology can respond to perturbations with rapid cell division, it is highly fragile to the invasion of mutant cells (Fig. 4C). Mutants with activating mutations (stronger gain) will accumulate due to their strong growth advantage resulting in rapid hyperplasia and clonal dominance compared with the DR topology, where these processes occur much more slowly. There is thus an inherent trade-off between the ability of a topology to reject mutants and its ability to support growth and recovery from large perturbations. This trade-off can be crucial to determine which topology would appear in specific physiological contexts.

# Discussion

The mechanisms that regulate tissue development and homeostasis are complex and involve many signalling, cell identity, and cell division pathways. To make progress in understanding their dynamics, it is important to identify simplifying operating principles. Here, motivated by a range of experimental phenomenology, we developed such an approach by studying the interaction between the cell cycle and cell fate decisions. We propose that regulation of cell fate by G1 lengthening occurs through a saddle-node bifurcation that, through cell–cell interactions, is self-tuned to the vicinity of its critical point. This mechanism provides unique benefits for maintaining a robust set point while leading to the rejection of mutant cells. This mechanism also presents trade-offs that may underpin major physiological changes in development, homeostasis, and ageing.

While the model is based on a well-established molecular mechanism for triggering the G1-S transition (the Rb/E2F bistable switch), its underpinnings are generic and based on universal properties of systems undergoing a saddle-node bifurcation. The effects of mitogens and factors such as cyclins and cell cycle inhibitors are all captured by an effective control parameter that is self-tuned through cell–cell interactions. From the point of view of modelling, this allows us to capture the dynamics of a complex regulatory network by a minimal effective theory. The critical nature of the mechanism also provides the system with a great deal of redundancy, allowing it to integrate many growth signals into a single effective control parameter. Moreover, it allows the circuit to function properly even upon the deletion of specific factors. The mechanism also has significant advantages for signal processing, as it allows for the temporal integration of mitogenic signals over prolonged timescales, thus providing a useful decision-making mechanism in noisy environments (Graf and Machta, 2024; Karin et al, 2023; Koch et al, 2024; Mora and Bialek, 2011; Nandan and Koseska, 2023; Stanoev et al, 2020).

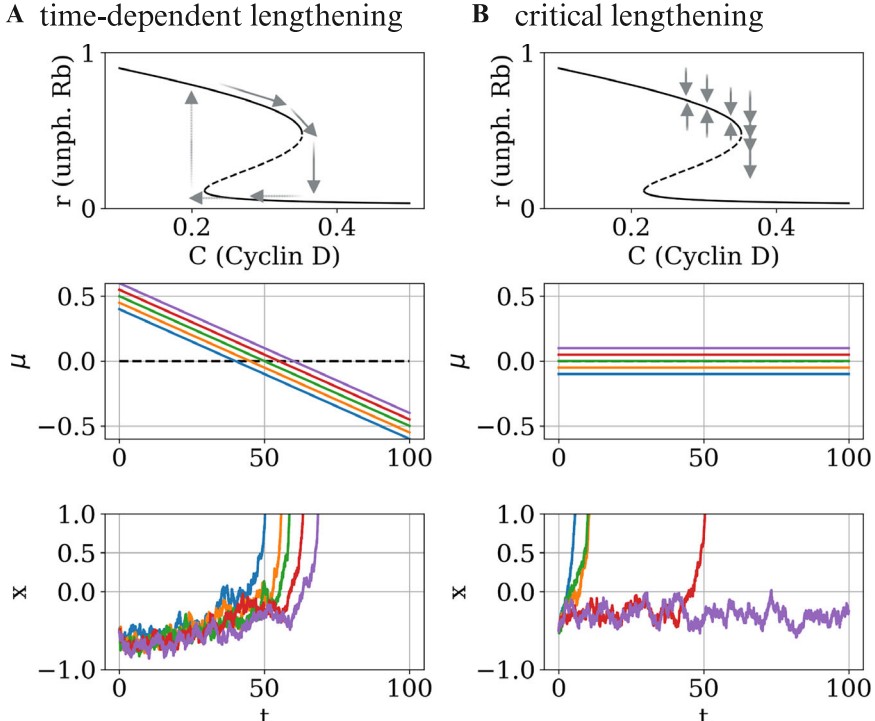

**Figure 5. Comparison of critical versus time-dependent G1 lengthening.**

(A) Time-dependent G1 lengthening is associated with a strong time-dependence of $\mu$, the distance from the critical point. (B) By contrast, in the critical lengthening regime, $\mu$ is held relatively constant over long timescales. The lower panels show a comparison of the two mechanisms, respectively, illustrated by simulations of Eq. (2). The results demonstrate the divergence of G1 lengths associated with the critical lengthening regime, which underlies rapid responses to perturbations (RD topology) and rejection of mis-sensing mutants (DR topology). The dynamics were simulated with a noise strength of $\xi = 0.1$.

The critical lengthening mechanism differs from previously proposed G1 lengthening mechanisms, which rely on time-dependent changes in circuit parameters (Schmoller et al, 2015; Tyson and Novak, 2014; Zatulovskiy et al, 2020). In these models, as G1 progresses, circuit parameters gradually shift, eventually causing the system to cross the critical threshold required for the G1-S transition. Both frameworks can be captured by Eq. (2) by introducing a time-dependent distance from bifurcation, $\mu(t)$ (Fig. 5). However, the two models exhibit fundamentally different behaviours and make distinct predictions. A strong time-dependence of $\mu$ (Fig. 5A) would imply weak sensitivity to input signals and to cellular variability, and thus provides an ideal mechanism for synchronous and oscillatory cell divisions (Ferrell Jr et al, 2009). This stands in stark contrast to the properties of the critical lengthening mechanism (Fig. 5B), which are highly sensitive to cellular variation. Notably, the time-dependent model does not account for experimental observations on the strong sensitivity of G1 length. The experimental observations of Rb protein decreasing in early G1, followed by a relative plateau (Zhang et al, 2024), suggest that both strategies may be employed by cells. Low cell–cell variability may dominate for short G1 lengths due to the time-dependence of the critical point, while critical lengthening and its associated benefits (rapid responses for RD topology, mutant rejection for DR topology) may prevail in slow-cycling cellular populations.

Our model reveals a fundamental trade-off between the RD and DR topologies. The RD topology, while enabling rapid population recovery from low cell numbers, is inherently fragile to invasion by "cheater" mutants with shorter G1 phases. Conversely, the DR topology is robust against such mutations, as it preferentially removes faster-cycling cells through differentiation before they can self-renew. However, this robustness comes at a significant cos:, namely a severely limited ability to recover from population bottlenecks.

These inherent fragilities may be overcome by complementary mechanisms. For the RD topology, for instance, biphasic regulation, where growth becomes negative at very fast division rates, could prevent mutant accumulation, albeit at the risk of population collapse (Karin and Alon, 2017). For the DR topology, recovery from low population abundances could be boosted by periodic signals that force cells into the cycle. Such a mechanism is observed in spermatogenesis, where the seminiferous cycle provides a periodic signal that drives stem cell division approximately every 10 days. Following division, these stem cells appear to enter a transient state that is more susceptible to differentiation but can be rescued by self-renewal factors present at that stage of the cycle (Nakagawa et al, 2021). This process may potentially act as a variation of the DR topology: it prevents runaway "cheater" mutants that escape the seminiferous cycle's control, while the periodic nature of the signal ensures the population can recover from large perturbations, as has been observed experimentally (Kitadate et al, 2019).

At the population level, the dynamics of the model leads to a self-renewing steady-state associated with fine-tuning of the

underlying molecular rates near the critical point. As such, it relates to previously demonstrated mechanisms of self-tuning to a critical point by competition in the context of transgenerational epigenetic inheritance of gene silencing and immune cell survival (Karin et al, 2023; Simons and Karin, 2024). However, unlike previous studies, this mechanism is associated with the steady-state levels of a single self-renewing population, rather than representing a balance between arrival and removal of competing entities. As in earlier work, the key to self-tuning is the sharp sensitivity of the delay around the critical point, which ensures that any extended division time must occur in its vicinity. This sensitivity amplifies variations between cells, and enables the circuit features of rapid responses and efficient rejection of mutant cells.

The unique sensitivity of G1 length at the bifurcation point allows tissues to have both a very low steady-state turnover and high responsiveness to perturbations. One system where this may be especially crucial is the regulation of beta cell mass in the pancreas. Beta cells have exceptionally low turnover in adults (less than 0.1% per day) (Saisho et al, 2013; Teta et al, 2005), but can increase their replication rate rapidly upon the induction of insulin resistance (Sakaguchi et al, 2017), allowing for physiological adaptation (Karin et al, 2016). This rapid response can be interpreted using the current model. The primary mitogen for beta cells is glucose, which progresses the G1-S transition by upregulating cyclin D expression (Salpeter et al, 2011). We, therefore, propose the following model for beta cell dynamics: when glucose levels are low, the cell cycle is extended in the stable regime, and the beta cell mass is quiescent. An elevation in glucose levels transitions the system to the unstable regime, which is associated with rapid cell divisions, until glucose levels are again lowered, and the system returns to the stable regime. In this system, there is biphasic regulation, where rapidly dividing cells are eliminated, allowing for mutant cell rejection, but conferring risk to glucotoxicity and diabetes (Karin and Alon, 2017). This mechanism is predicted to exploit the sharp sensitivity around the critical point to generate sharp transitions between quiescence and expansion and to remove mis-sensing mutants.

Similar mechanisms may underpin quiescence in adult tissues dominated by the DR topology. Prolonged G1 is a "soft" direction as cells have very slow growth rates. The system may thus transition to this state, only to be reactivated when cues for differentiation transition it back to the unstable phase. The RD topology, on the other hand, does not generate a similar fraction of quiescent cells as, in this case, the tail of the distribution is associated with cells destined for removal. The strong sensitivity of the dynamics around the critical point allows the population to induce quiescence in specific subpopulations, such as DNA-damaged cells, while retaining normal growth for other cells, in line with experimental observations (Barr et al, 2017).

While our analysis was presented in a relatively simple homeostatic setting, it can be readily extended to other settings, including the regulation of cellular composition in growing cell populations, as occurs, for example, in the pre-implantation embryo (Saiz et al, 2020). This case is compatible, for example, with feedback from the concentration of one cell type to another. In homeostatic settings, there may also be feedback on the size of the pool of differentiated cells, rather than on the size of the self-renewing compartment. Finally, it is important to note that, while our theory focused on G1 lengthening, its main tenets apply to any

regulation based on temporal modulation of transient states, including, for example, the modulation of other cell cycle stages or of states unrelated to the cell cycle.

In addition to modelling biological systems, understanding mechanisms for the regulation of cellular dynamics is crucial for the application of synthetic biology in health and industrial settings, where robust control and mutant cell elimination are central challenges (Glass et al, 2024; Ma et al, 2022; Robinson et al, 2021; You et al, 2004). We propose that, due to its prevalence in natural systems and the clear functional benefits analysed in this study, the regulation of cell fate decisions by temporal modulation may provide an effective design principle for cell population control.

## Methods

### Population dynamics

We consider a population of (stem) cells with identical cell-cycle duration $T$. Let $P_R(T)$ denote the probability that a division, occurring after duration $T$, results in self-renewal, and $P_D(T)$ the probability of differentiation. (We assume, for simplicity of analysis, symmetric cell fates; our conclusions can be straightforwardly generalised to asymmetric fates). The net expected change in population size per cycle is

$$\Delta N = (P_R(T) - P_D(T))N \equiv \theta(T)\,N,$$

where we define

$$\theta(T) = P_R(T) - P_D(T) = 2P_R(T) - 1. \qquad (8)$$

In discrete time, after each cycle of length $T$,

$$N_{k+1} = [1 + \theta(T)]\,N_k.$$

The corresponding continuous-time growth rate is

$$\frac{\dot{N}}{N} = \frac{1}{T}\ln[1 + \theta(T)].$$

Near a steady state where $\theta(T_s) = 0$, the logarithm can be linearized, yielding

$$\frac{\dot{N}}{N} \approx \frac{\theta(T)}{T}. \qquad (9)$$

The function $\theta(T)$ captures the net self-renewal bias and we assume that it has a single intersection point with the $T$ axis (and therefore a single fixed point), at least within the physiologically relevant range of $T$. This feedback is mediated by the control parameter, $C$, such that $C = f(N)$ and $T = \Phi(C)$, where $\Phi(C)$ is the function that maps the control parameter to the G1 duration. A steady-state is achieved at a population size $N_s$ where $\dot{N} = 0$, which requires that $\theta(T_s) = 0$ for some steady-state G1 duration $T_s$.

Consider a small perturbation, $\delta N$, from the steady-state, such that $N = N_s + \delta N$. This perturbation propagates through the feedback loop:

- The change in population size alters the control parameter $C$:

$$\delta C \approx \left.\frac{df}{dN}\right|_{N_s} \delta N$$

- The change in $C$ alters the G1 duration $T$:

$$\delta T \approx \left.\frac{d\Phi}{dC}\right|_{C_s} \delta C$$

- The change in $T$ alters the net growth bias $\theta(T)$:

$$\delta\theta \approx \left.\frac{d\theta}{dT}\right|_{T_s} \delta T$$

Combining these steps, the response of the growth bias to the initial population perturbation is:

$$\delta\theta \approx \left(\left.\frac{d\theta}{dT}\right|_{T_s}\left.\frac{d\Phi}{dC}\right|_{C_s}\left.\frac{df}{dN}\right|_{N_s}\right)\delta N \tag{10}$$

For the set-point $N_s$ to be stable, a positive perturbation ($\delta N > 0$) must induce a negative growth rate ($\dot{N} < 0$, which implies $\theta < 0$ and thus $\delta\theta < 0$). This requires the term in the parenthesis to be negative:

$$\left.\frac{d\theta}{dT}\right|_{T_s}\left.\frac{d\Phi}{dC}\right|_{C_s}\left.\frac{df}{dN}\right|_{N_s} < 0 \tag{11}$$

From the fundamental properties of the saddle-node bifurcation (Fig. 1C), an increase in the control parameter $C$ (e.g., mitogen level) shortens the G1 duration $T$. Therefore, the derivative $\frac{d\Phi}{dC}$ is negative. For the entire expression to be negative, the product of the remaining two derivatives must be positive:

$$\left.\frac{d\theta}{dT}\right|_{T_s}\left.\frac{df}{dN}\right|_{N_s} > 0 \tag{12}$$

This condition is only satisfied if both derivatives have the same sign, leading to the stability criterion presented in the main text:

$$\text{sign}\left(\frac{d\theta}{dT}\right) = \text{sign}\left(\frac{df}{dN}\right) \tag{13}$$

## Mechanistic examples of population feedback

To provide a more concrete biological grounding for the feedback function $C = f(N)$, we consider a simplified model based on a diffusible ligand that illustrates how both the Renewal precedes Differentiation (RD) and Differentiation precedes Renewal (DR) topologies can arise.

Consider a ligand, $l$, that is secreted at a constant rate, $p$, by the surrounding niche and is consumed or degraded by the stem cells, $N$, at a per-capita rate $k_{deg}$. The dynamics of the ligand concentration can be described as:

$$\frac{dl}{dt} = p - k_{deg}Nl \tag{14}$$

Assuming the ligand dynamics are fast relative to cell division, the concentration reaches a steady-state ($dl/dt = 0$) given by $l_{ss} = p/(k_{deg}N)$. The resulting feedback topology now depends entirely on whether the ligand acts as a mitogen or an inhibitor.

**Case 1: ligand is a mitogen (RD topology)**. If the ligand promotes cell cycle progression, the control parameter $C$ will be proportional to the ligand concentration, $C = \alpha l_{ss}$. The feedback function is therefore:

$$C = f(N) = \frac{\alpha p}{k_{deg}N} \tag{15}$$

The derivative of this function, $df/dN$, is negative. Stability requires that $d\theta/dT < 0$, meaning a longer G1 phase is associated with differentiation. This corresponds to the RD topology.

**Case 2: Ligand is an inhibitor (DR topology)**. If the ligand inhibits cell cycle progression (e.g., represses Cyclin D), the control parameter $C$ will be inversely proportional to the ligand concentration, $C = \beta/l_{ss}$. The feedback function becomes:

$$C = f(N) = \frac{\beta k_{deg}N}{p} \tag{16}$$

The derivative of this function, $df/dN$, is positive. Stability now requires that $d\theta/dT > 0$, meaning a longer G1 phase is associated with self-renewal. This corresponds to the DR topology.

**The role of timescales and the quasi-steady-state assumption**. Our model relies on a separation of timescales. The intracellular dynamics of the Rb/E2F network operate on a timescale of minutes to hours. In contrast, population dynamics, characterized by cell division and changes in the total cell number $N$, occur on a much slower timescale of many hours to days.

The environmental feedback, mediated by the ligand $l$, must operate on a timescale that is fast relative to the timescale of cell division. This is the justification for the quasi-steady-state assumption ($dl/dt \approx 0$) used above. If the ligand dynamics were slower than cell division, a cell would be responding to an outdated environmental signal that reflects a past population size. Such a significant delay would introduce instability into the feedback loop, potentially leading to large oscillations around the homeostatic set-point rather than stable regulation.

**Generality of the feedback mechanism: control by differentiated cells**. In the main text, we considered a feedback loop where the undifferentiated stem cell population, $N$, directly regulates its own size. However, it is also plausible that feedback is mediated by the differentiated cell population, $D$. Here, we show that our model's conclusions are robust to this alternative regulatory structure.

Let us consider a two-compartment model. Stem cells, $N$, divide with a G1 duration $T$. Each division produces, on average, $(1 - \theta(T))$ differentiated cells and $(1 + \theta(T))$ stem cells. The differentiated cells, $D$, are removed (e.g., die) at a fixed rate, $\gamma$. The system dynamics can be described as:

$$\frac{dN}{dt} = \frac{N}{T}\theta(T) \tag{17}$$

$$\frac{dD}{dt} = \frac{N}{T}(1 - \theta(T)) - \gamma D \tag{18}$$

Now, let's assume that the differentiated cells, $D$, secrete a ligand, $l$, at a rate $p_D$, which is then degraded. At steady-state, the ligand concentration is proportional to the size of the differentiated population, $l_{ss} \propto D$. If this ligand controls the stem cell cycle parameter $C$, then the feedback function becomes $C = f(D)$.

For the system to reach a homeostatic steady-state, both populations must be constant ($\dot{N} = 0$ and $\dot{D} = 0$). The condition $\dot{N} = 0$ requires that $\theta(T_s) = 0$ at some steady-state G1 length $T_s$. The condition $\dot{D} = 0$ then implies that, at this steady-state, $N_s/T_s = \gamma D_s$. This shows a stable coupling between the two population sizes. As in the single-compartment model, any homeostatic state that relies on a prolonged G1 duration, $T_s$, can only be achieved if the system is tuned to the vicinity of the critical point, where G1 length is exquisitely sensitive to the control parameter.

Crucially, the stability analysis and the emergence of the RD and DR topologies remain unchanged. The feedback loop now passes through the differentiated compartment, but the logic is identical: a perturbation in $N$ affects $D$, which in turn affects $C$ and feeds back on $N$. For example, if the ligand is a mitogen ($C \propto D$), an increase in $N$ leads to an increase in $D$, which increases $C$ and shortens $T$. This corresponds to $df/dN > 0$, requiring the DR topology for stability.

Therefore, the core mechanism of self-tuning to criticality and the resultant trade-offs between rapid growth and mutant rejection are general principles that hold regardless of whether the feedback is mediated directly by stem cells or indirectly by their differentiated progeny.

## Simulation methodology and parameters

To investigate the dynamics of the cell population (Figs. 3 and 4), we implemented a custom agent-based simulation in Python using the NumPy library for efficient vectorization. Each cell in the population is an independent agent, characterized by several key attributes: its status as an active member of the population, its age within the G1 phase, the current concentration of active Rb protein, a stochastically assigned differentiation commitment time, and an individual sensitivity factor to feedback signals (used for mutation simulations).

The simulation proceeds in discrete time steps, $\Delta t$. At each step, the following updates are performed for all active cells:

1. Feedback calculation: The total number of live cells, $N$, is counted. This number is used to calculate the global Cyclin D concentration, $C$, according to the feedback function $C = f(N)$ specific to the RD or DR topology. For simulations with mutations, this is modified to $C = f(\alpha_i N)$, where $\alpha_i$ is the individual cell's sensitivity.
2. Rb dynamics: The concentration of active Rb, $r_i$, for each cell is updated according to the stochastic differential equation (Eq. 1 in the main text), using a forward Euler-Maruyama step.
3. Age update: The age of each cell is incremented: $age_i \leftarrow age_i + \Delta t$.
4. Division check: A cell is marked for division if its active Rb level drops below the G1/S threshold ($r_i < r_{thresh}$). An additional, slow, baseline division rate is included to account for stochastic, unregulated cell cycle entry, with a probability of $\Delta t/T_{baseline}$ per time step.
5. Fate determination: The simulation implements a "competing risks" model, which gives rise to the function $\theta(T)$. A cell that is marked for division is checked against its differentiation commitment time.

- In the RD topology, a cell that divides before its age exceeds its commitment time undergoes self-renewal. If its age exceeds its commitment time, it is removed from the population (differentiates).
- In the DR topology, this logic is inverted. Division before the commitment time leads to differentiation, while division after leads to self-renewal.

The precise choice of the fate function $\theta(T)$ is not critical for our conclusions, as long as the steady-state balance ($\theta(T_s) = 0$) occurs at a sufficiently large $T_s$ that requires critical lengthening.

6. Cell division and mutation: For each self-renewing division, the parent cell's state is reset (age = 0, r = $r_0$, a new commitment time is drawn), and a new daughter cell is created with an identical state in an empty slot in the array. During mutation simulations, the daughter cell's sensitivity parameter, $\alpha_i$, has a small probability of being altered by multiplication with a random number drawn from a log-normal distribution.

The parameters used in the simulations are listed in Table 1. Values were chosen to be consistent with the known qualitative behaviour of the Rb/E2F system and typical cell cycle timescales.

In our model for the dynamics of active Rb, $r$, the phosphorylation rate includes a term $\gamma_P \propto 1/(1 + (r/k_2)^2)$. This term, which has the form of a Hill function with a Hill coefficient of

**Table 1.   Simulation parameters**

| Parameter | Value | Description |
|---|---|---|
| *Rb/E2F model parameters (Eq. 1)* | | |
| $\gamma_D$ | 1.0 | Degradation/dilution rate of active Rb, scaled to 1. |
| $V$ | 100 | Maximal rate of Rb phosphorylation. |
| $k_1$ | 1.0 | Michaelis-Menten constant for Cyclin D saturation. |
| $k_2$ | 0.1 | Michaelis-Menten constant for Rb self-regulation; sets the half-max point for the feedback. |
| $\sigma$ | 0.3 | Noise intensity for the stochastic term. |
| $r_0$ | 4.0 | Initial concentration of active Rb for new cells. |
| $r_{thresh}$ | 0.05 | G1/S transition threshold for active Rb. |
| $C_{crit}$ | 0.35 | Critical Cyclin D concentration for the bifurcation. |
| *Population simulation parameters (Figs. 3 and 4)* | | |
| $T_{baseline}$ | 720 h (30 days) | Mean time for baseline (unregulated) cell division. |
| $\tau_d$ | 20 h | Mean timescale for the stochastic differentiation timer. |
| $N_{st}$ | 100 | Target population set-point (large population). |
| $N_{st}$ | 10 | Target population set-point (small population). |
| $\Delta t$ | 0.1 h | Simulation time step. |
| *Mutation simulation parameters (Fig. 4)* | | |
| $p_m$ | 0.001 | Probability of mutation per division. |
| $\sigma_{mut}$ | 0.5 | Log-standard deviation for the multiplicative mutation effect. |
| $\alpha_{min}$ | 0.03 | Minimum allowed sensitivity value for mutants. |

$n = 2$, is chosen to phenomenologically capture the cooperative positive feedback loop mediated by the E2F/CycE cascade. Active Rb represses E2F, which in turn promotes the expression of Cyclin E. Cyclin E then phosphorylates and inactivates Rb. Such positive feedback loops are known to generate bistability, and the use of a Hill coefficient greater than 1 is a standard and minimal way to model the required nonlinearity for creating a bistable switch (Yao et al, 2008). This precise choice is not important for our conclusions, which depend on the local behaviour near the bifurcation point and therefore generalise to any model that has bistability and can undergo a saddle-node bifurcation.

To compare our model with the experimental data from Jang et al (2019), we simulated G1 exit times for populations of non-interacting cells under different conditions. For these simulations, the Cyclin D concentration, $C$, for each cell was drawn from a normal distribution, $C \sim \mathcal{N}(C_m, C_s^2)$. The standard deviation was fixed at $C_s = 0.05$. The mean, $C_m$, was set for each experimental condition based on the reported relative Wnt activity: mTeSR1 ($C_m = 0.5$), E8+IWP2 ($C_m = 0.6$), E8 ($C_m = 0.7$), and mTeSR1+Wnt ($C_m = 1.5$).

The full simulation code is openly available at https://github.com/karin-lab/cellcycle.

### Simulation of ghost dynamics (Fig. 2)

To simulate the dynamics of the intermediate state, we ran a simulation of $N = 50{,}000$ non-interacting cells for $t_{max} = 100$ h with a time step of $\Delta t = 0.05$ h. Cyclin D activity for each cell, $C_i$, was drawn from a normal distribution, $\mathcal{N}(C_m, C_s^2)$, with a mean ($C_m = C_{crit} + 0.008$) and standard deviation ($C_s = 0.004$).

### Data processing for trajectories and histograms

The Rb-E2F switch activity was proxied by the transformation $1 - \tanh(r)$, which bounds the activity between 0 and 1. For each simulated cell, we identified three time points based on when its activity crossed specific thresholds:

- $t_{active}$: Time of crossing the "active" threshold (0.4).
- $t_{high}$: Time of crossing the intermediate threshold (0.7).
- $t_{S-phase}$: Time of crossing the S-phase entry threshold ($1 - \tanh(r_{thresh})$).

From these time points, we calculated the total G1 length ($t_{S-phase} - t_{active}$) and the durations of the two phases shown in the histograms in Panel C.

For the trajectory plot in Panel B, the cells were binned according to their total G1 length. Within each bin, all individual trajectories were aligned by setting their S-phase entry time to $t = 0$. These aligned trajectories were then interpolated onto a common time axis to calculate the average trajectory for that bin, which is plotted as a thick coloured line. A single, randomly selected trajectory from each bin is also shown as a lighter, thinner line to represent typical cell-to-cell variability.

To simulate the perturbation experiment, we ran a simulation of $N = 100$ cells for $t_{max} = 80$ h. All cells started with a constant Cyclin D concentration of $C = C_{crit}$. At $t_{perturb} = 15$ h, the concentration was suddenly dropped by a magnitude of $\Delta C = -0.1$.

After the perturbation, cells were classified into two groups based on their subsequent behaviour. A cell was classified as "proceeds to S" if its activity crossed the S-phase entry threshold at any point after the perturbation. A cell was classified as "reverts to quiescence" if it did not enter S-phase and its activity level was high (>0.5) just before the perturbation but then dropped to a low level (<0.4 on average) by the end of the simulation. Five representative trajectories from each class were plotted.

### Generalized model for G1 lengthening as a noisy saddle-node bifurcation

In the main text, we considered a minimal model of the G1-S transition based on the concentration of active Rb protein. More generally, we can model the length of G1 using the following general stochastic dynamics,

$$d\mathbf{v} = f(\mathbf{v}, C)\, dt + s(\mathbf{v})\, dW, \tag{19}$$

where $\mathbf{v} = v_1 \ldots v_K$ represents the state of a gene regulatory network (GRN), $f = f_1 \ldots, f_K$ defines the dynamics of the GRN, $s = s_1, \ldots, s_K$ defines the state-dependent amplitude of the noise, $W$ is a $K$-dimensional Wiener process, and $C$ is a control parameter for a saddle-node bifurcation that destabilizes the G1 state, denoted $\mathbf{v}_{G1}$. While $f$ can correspond to the dynamics of the Rb/E2F/CycE network and $C$ to the concentration of CycD, the general analysis does not make specific assumptions about the number of components or dynamics of the GRN, allowing us to make general conclusions that extend to other possible molecular mechanisms. In this sense, Eq. [1] in the main text is a specific example of Eq. (19).

In an experimental setting, there may be slight differences in biochemical parameters or sensed environments between individual cells. In our model, we can capture this by indexing the GRN of each cell as $\mathbf{v}_i$ (with $i = 1, 2, \ldots$) and setting $C_i$ as a random variable. The dynamics of a cell population are then given by

$$d\mathbf{v}_i = f(\mathbf{v}_i, C_i)dt + s(\mathbf{v}_i)dW_i. \tag{20}$$

Variation in the control parameters $C_i$, as well as noise, cause cell-to-cell differences in the length of G1. We can also think of variation in $C_i$ as fluctuations on the timescale of cell growth and division, while $W$ captures fluctuations on a faster timescale. In the vicinity of the critical point, the dynamics of Eq. (20) are captured by the stochastic normal form, Eq. (2) in the main text, where the GRN coordinates and time are rescaled to $x$ and $\hat{t}$, while $\xi$ corresponds to the fluctuations near the critical point, and $\mu_i$ is the (rescaled) distance from bifurcation of cell $i$. More formally, $\mu_i \propto C_i - C_{crit}$. The duration of G1 is then given by (Hathcock and Sethna, 2021b)

$$T_i = 2^{1/3}\pi^2\xi^{-2/3}\left(\text{Ai}^2[-2^{2/3}\xi^{-4/3}\mu_i] + \text{Bi}^2[-2^{2/3}\xi^{-4/3}\mu_i]\right), \tag{21}$$

where Ai, Bi are Airy functions of the first and second kind, respectively. There is a steep dependence of $T$ on the bifurcation parameters, so that $T_i$ increases super-exponentially as $\mu_i$ decreases (and thus as $C_i$ drops below the critical value).

## Data availability

This study includes no data deposited in external repositories. The full analysis and simulation code are available at https://github.com/karin-lab/cellcycle.

The source data of this paper are collected in the following database record: biostudies:S-SCDT-10_1038-S44320-025-00164-8.

## Peer review information

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

## Acknowledgements

We wish to thank Alexis Barr for helpful discussions and critical reading of the manuscript. BDS acknowledges the support of the Royal Society through an EP Abraham Research Professorship (RSRP/R/231004). For the purpose of open access, the authors have applied a Creative Commons Attribution (CC BY) licence to any Author Accepted Manuscript version arising from this submission. OK acknowledges funding from the Imperial College London Open Access Fund. Figures were created with BioRender.com.

## Author contributions

**Benjamin D Simons**: Writing—review and editing. **Omer Karin**: Conceptualization; Formal analysis; Investigation; Visualization; Methodology; Writing—original draft.

Source data underlying figure panels in this paper may have individual authorship assigned. Where available, figure panel/source data authorship is listed in the following database record: biostudies:S-SCDT-10_1038-S44320-025-00164-8.

## Disclosure and competing interests statement

The authors declare no competing interests.

# Expanded View Figures

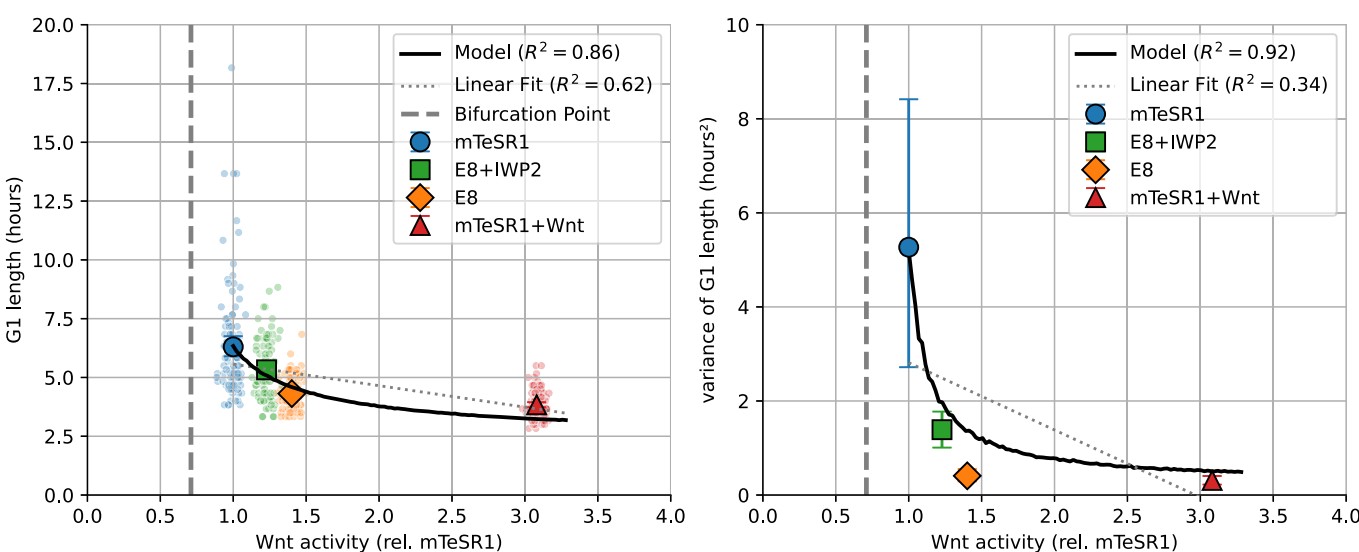

**Figure EV1. Quantitative comparison of G1 lengthening models.**

Comparison of experimental data ($n = 114$ for mTeSR1, $n = 105$ for E8+IWP2, $n = 104$ for mTeSR1+Wnt, and $n = 112$ for E8) from Jang et al (2019) with model predictions for the mean (left) and variance (right) of G1 length in hESCs under different Wnt signalling conditions. The model, based on a noisy saddle-node bifurcation (solid black line), captures the dependency of both the mean ($R^2 = 0.86$) and the variance ($R^2 = 0.92$) of G1 length on Wnt activity, outperforming linear regressions (dotted grey line, $R^2 = 0.62$ and $R^2 = 0.34$, respectively). Experimental data points show bootstrapped means (centre of error bars) with 95% confidence intervals. The dashed vertical line indicates the bifurcation point.

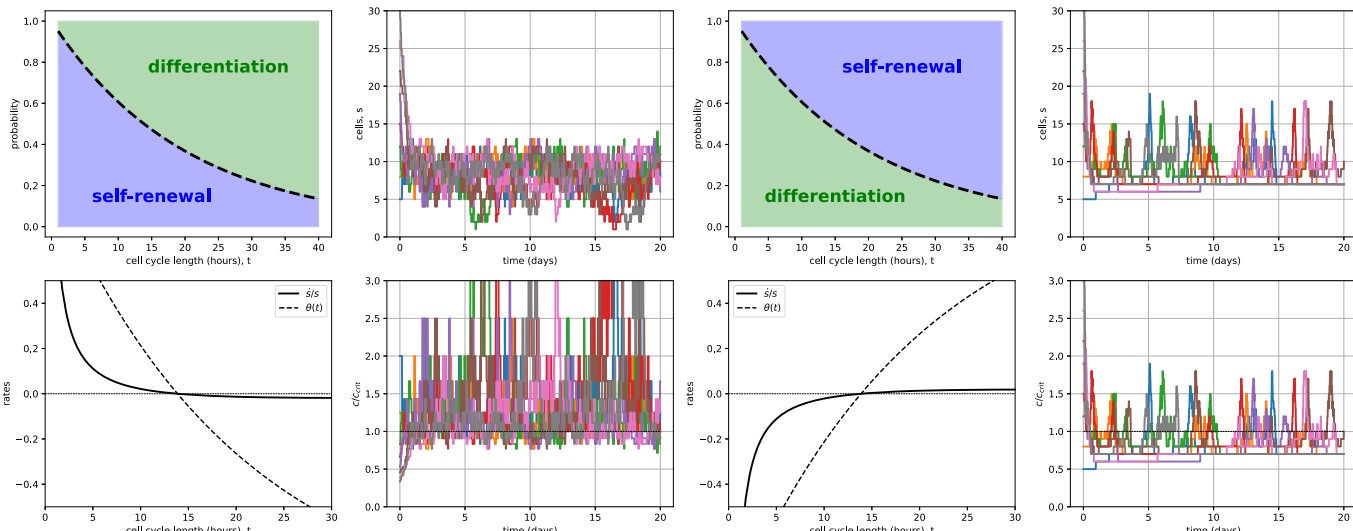

**Figure EV2.  Robust population control in small stem cell niches.**

To test the robustness of our proposed control mechanism, we simulated the population dynamics for both the RD (left) and DR (right) topologies in a small stem cell niche with a target set-point of $N = 10$ cells. The simulations show that even in the presence of significant demographic fluctuations inherent to small populations, the negative feedback loop effectively maintains the population size around the set-point. In both topologies, the system consistently self-tunes to the vicinity of the critical point (bottom right panels), demonstrating that the control mechanism is robust and does not require a large population size to function.

