## [Peer Review File · Molecular Systems Biology]

Cell cycle criticality as a mechanism for robust cell population control

Benjamin Simons and Omer Karin

Corresponding author(s): Omer Karin (o.karin@ic.ac.uk)

Review Timeline:

Submission Date:	10th Jun 25
Editorial Decision:	29th Jul 25
Revision Received:	21st Aug 25
Editorial Decision:	25th Sep 25
Revision Received:	26th Sep 25
Accepted:	1st Oct 25

Editor: Jingyi Hou

Transaction Report:

29th Jul 2025

Manuscript Number: MSB-2025-13167

Title: Cell cycle criticality as a mechanism for robust cell population control

Author: Benjamin Simons

Omer Karin

Dear Dr Karin,

Thank you for submitting your manuscript to Molecular Systems Biology. First, I would like to apologize for the delay in the review process, which was due to the late arrival of the reviewers' reports. We have now received feedback from all four reviewers who agreed to evaluate your work. As you will see from their comments below, the reviewers recognize the interest and potential significance of your study. However, they have also raised a number of substantial concerns that will need to be addressed through a major revision.

In particular, Reviewers #2 and #3 have expressed concerns regarding several assumptions used in your model. Additionally, Reviewer #3 questioned the overall relevance of the work to real biological systems. These are critical issues that require careful consideration, and we encourage you to strengthen the biological grounding of your study accordingly.

Please note that due to formatting issues with mathematical equations in the online system, Reviewer #1 has submitted their comments as a separate Word document, which is attached for your reference.

All other reviewer comments should be addressed comprehensively as well. As per our editorial policy, we generally allow only one round of major revision, so it is important that your responses be as thorough and complete as possible.

On a more editorial level, we would ask you to address the following issues:

- Please provide a .docx formatted version of the manuscript text (including legends for main figures, EV figures and tables). Please make sure that the changes are highlighted to be clearly visible.
- Please provide individual production quality figure files as .eps, .tif, .jpg (one file per figure).
- Please provide a .docx formatted letter INCLUDING the reviewers' reports and your detailed point-by-point responses to their comments. As part of the EMBO Press transparent editorial process, the point-by-point response is part of the Review Process File (RPF), which will be published alongside your paper.
- Please note that all corresponding authors are required to supply an ORCID ID for their name upon submission of a revised manuscript.
- We replaced Supplementary Information with Expanded View (EV) Figures and Tables that are collapsible/expandable online (see examples in <http://msb.embopress.org/content/11/6/812>). A maximum of 5 EV Figures can be typeset. EV Figures should be cited as 'Figure EV1, Figure EV2' etc... in the text and their respective legends should be included in the main text after the legends of regular figures.

Additional Tables/Datasets should be labeled and referred to as Table EV1, Dataset EV1, etc. Legends have to be provided in a separate tab in case of .xls files. Alternatively, the legend can be supplied as a separate text file (README) and zipped together with the Table/Dataset file.

For the figures and tables that you do NOT wish to display as Expanded View figures, they should be bundled together with their legends in a single PDF file called *Appendix*, which should start with a short Table of Content. Each legend should be below the corresponding Figure/Table in the Appendix. Appendix figures and tables should be referred to in the main text as: "Appendix Figure S1, Appendix Figure S2, Appendix Table S1" etc. See detailed instructions regarding expanded view here: <https://www.embopress.org/page/journal/17444292/authorguide#expandedview>.

- Before submitting your revision, primary datasets (and computer code, where appropriate) produced in this study need to be deposited in an appropriate public database (see <http://msb.embopress.org/authorguide-dataavailability>).

<https://www.embopress.org/page/journal/17444292/authorguide#dataavailability>).

The accession numbers and database should be listed in a formal "Data Availability" section (placed after Materials & Method) that follows the model below (see also <https://www.embopress.org/page/journal/17444292/authorguide#dataavailability>). Please note that the Data Availability Section is restricted to new primary data that are part of this study.

Data availability

Additional information on source data and instruction on how to label the files are available

- Our journal encourages inclusion of *data citations in the reference list* to directly cite datasets that were re-used and obtained from public databases. Data citations in the article text are distinct from normal bibliographical citations and should directly link to the database records from which the data can be accessed. In the main text, data citations are formatted as follows: "Data ref: Smith et al, 2001". In the Reference list, data citations must be labeled with "[DATASET]". A data reference must provide the database name, accession number/identifiers and a resolvable link to the landing page from which the data can be accessed at the end of the reference. Further instructions are available at .

- We updated our journal's competing interests policy in January 2022 and request authors to consider both actual and perceived competing interests. Please review the policy <https://www.embopress.org/competing-interests> and update your competing interests if necessary.
Please use the heading "Disclosure statement and competing interests".

- All Materials and Methods need to be described in the main text using our 'Structured Methods' format. According to this format, the Methods section includes a Reagents and Tools Table (listing key reagents, experimental models, software and relevant equipment and including their sources and relevant identifiers) followed by a Methods and Protocols section describing the methods, ideally using a step-by-step protocol format. The aim is to facilitate adoption of the methodologies across labs.

Please download and fill our Reagents and Tools Table template (.docx), which you can find in our author guidelines:

<https://www.embopress.org/page/journal/17444292/authorguide#structuredmethods>.

-Regarding data quantification:

Please ensure to specify the name of the statistical test used to generate error bars and P values, the number (n) of independent experiments (please specify technical or biological replicates) underlying each data point and the test used to calculate p-values in each figure legend. Discussion of statistical methodology can be reported in the materials and methods section, but figure legends should contain a basic description of n, P and the test applied.

Graphs must include a description of the bars and the error bars (s.d., s.e.m.).

- Please provide a "standfirst text" summarizing the study in one or two sentences (approximately 250 characters, including space), three to four "bullet points" highlighting the main findings and a "synopsis image" (550px width and 400-600 px height, PNG format) to highlight the paper on our homepage.

Here are a couple of examples:

<https://www.embopress.org/doi/10.15252/msb.20199356>

<https://www.embopress.org/doi/10.15252/msb.20209475>

<https://www.embopress.org/doi/10.15252/msb.209495>

When you resubmit your manuscript, please download our CHECKLIST (<https://www.embopress.org/pb-assets/embo-site/EMBO%20Press%20Author%20Checklist-1642513524327.xlsx>) and include the completed form in your submission.

Please note that the Author Checklist will be published alongside the paper as part of the transparent process (<https://www.embopress.org/page/journal/17444292/authorguide#transparentprocess>).

If you feel you can satisfactorily deal with these points and those listed by the referees, you may wish to submit a revised version of your manuscript. Please attach a covering letter giving details of the way in which you have handled each of the points raised by the referees. A revised manuscript will be once again subject to review and you probably understand that we can give you no guarantee at this stage that the eventual outcome will be favorable.

I look forward to receiving your manuscript soon.

Kind regards,
Jingyi

Jingyi Hou, PhD
Senior Editor
Molecular Systems Biology

We realize that it is difficult to revise to a specific deadline. In the interest of protecting the conceptual advance provided by the work, we recommend a revision within 3 months (27th Oct 2025). Please discuss the revision progress ahead of this time with the editor if you require more time to complete the revisions. Use the link below to submit your revision:

*** PLEASE NOTE *** As part of the EMBO Press transparent editorial process initiative (see our Editorial at <https://dx.doi.org/10.1038/msb.2010.72>), Molecular Systems Biology publishes online a Review Process File with each accepted manuscripts. This file will be published in conjunction with your paper and will include the anonymous referee reports, your point-by-point response and all pertinent correspondence relating to the manuscript. If you do NOT want this File to be published, please inform the editorial office at contact@molsystbiol.org within 14 days upon receipt of the present letter.

Reviewer #1:

I rated the 'validity of conclusions' medium because of some major revisions requested below.

My report copied below is difficult to read because all the mathematical notation was screwed up. I sent a copy of the Word document to the MSB email address, but I don't know if it got attached to this review.

Summary

The authors, Simons & Karin (S&K), put forward a mathematical model to understand how proliferating stem cells maintain an appropriate balance between self-renewal and differentiation, and, at the same time, eliminate deleterious mutants that outgrow wild-type cells. Although their model is quite abstract, it is based soundly on the experimentally established fact that the G1/S transition in mammalian cells is controlled by a saddle-node (SN) bifurcation. Their basic idea is that cell differentiation occurs from G1 phase of the cell cycle; however, as a control parameter C (e.g., the activity of CycD-dependent kinase) increases past a critical point, C_{crit} , the stable G1 state is eliminated at a SN bifurcation, and the cell makes an irreversible commitment to S-G2-M phase and cell division (i.e., self-renewal). From a mathematical point-of-view, all SN bifurcations 'look the same' and can be described by a simple stochastic differential equation

$$dx/dt = x^2 + \mu + \text{noise}$$

where x denotes the state of the cell cycle and $\mu = C - C_{crit}$ is the 'distance' of the control system from the SN bifurcation. Note: for $\mu = C - C_{crit} < 0$, $x(t)$ has two steady-state solutions, $x_- = -\sqrt{\mu}$ and $x_+ = +\sqrt{\mu}$, where $x_- < 0$ corresponds to the stable G1 state, and $x_+ > 0$ is an unstable saddle point. As μ passes through 0, the two steady states coalesce and disappear, and for $\mu > 0$, $x(t) \rightarrow +\infty$, which the authors interpret as an irreversible commitment to self-renewal. A generic property of SN bifurcations in the presence of noise, which is crucial to S&K's theory, is that, for C close to but less than C_{crit} , i.e., for $|\mu| \approx 0$, there is a finite time T when fluctuations may drive the control system off the G1 state (x_-), past the saddle point (x_+), and into DNA synthesis and division. Furthermore, transition time gets longer as the distance from criticality increases: $\log T \sim |\mu|^{3/2}$. Notice that $T = \Phi(C)$, with $\Phi(C_{crit}) = 0$ and $\partial\Phi/\partial C < 0$. The authors associate this time T to the duration of G1 phase and, hence, to the cell generation time (assuming the duration of S-G2-M is fixed and small, relative to T). In this case, the number $N(t)$ of undifferentiated stem cells is given by

$$dN/dt = N/T \theta(T)$$

where $\theta(T)$ specifies how the net rate of self-renewal depends on G1 duration. (Note: $\theta(T)$ can be < 0 because there are always some stem cells lost by cell death.) The tissue achieves balanced self-renewal ($N(t) = N_S = \text{constant}$, $dN/dt = 0$) for $\theta(T_S) = 0$ at some $T = T_S$. Now S&K point out that, since all finite T 's are found in the vicinity of $C = C_{crit}$, the steady-state population density is found in the vicinity of C_{crit} , i.e., $N_S = f^{-1}(C_{crit})$, where $C = f(N)$ denotes the dependence of the control parameter C on stem-cell density N . The authors point out that the fixed point of self-renewal, N_S , is stable if and only if the sign of $[\partial\theta/\partial T]$ is the same as the sign of $[\partial f/\partial N]$. Clearly there are two cases: (I) both signs are negative or (II) both signs are positive. The authors go on to

study these two cases in detail, showing that in case (I) self-renewal occurs rapidly, whereas case (II) is more resistant to being displaced by mutants that are biased toward self-renewal ('cheater' mutants).

General remarks

This manuscript presents a very elegant and (to my knowledge) novel approach to the question of tissue maintenance by self-renewing populations of stem cells, with particular emphasis on the important question of stability with respect to cheater mutations. The paper is well motivated and, for the most part, very clear. The figures are attractive and informative. The advances are primarily conceptual. The paper will be of interest to biological physicists, mathematical biologists, and cell biologists with a sophisticated understanding of dynamical systems theory (saddle-node bifurcations, stochastic differential equations, barrier crossing). The paper is very suitable for publication in *Molecular Systems Biology*.

Major points

1. "The authors point out that the fixed point of self-renewal, NS, is stable if and only if the sign of $[\partial\theta/\partial T]$ is the same as the sign of $[\partial f/\partial N]$." I am sure the authors are correct in this statement, but it is not obvious from their Eqs. (3-5). It took me some time to convince myself of the claim, and I had to introduce the function $T = \Phi(C)$, with $\Phi(C_{crit}) = 0$ and $\partial\Phi/\partial C < 0$ in order to connect the function $f(N)$ to $\theta(T)$. Because Eq. (6) is central to the entire argument, the authors need to provide a convincing demonstration of this claim.
2. In the Discussion (Fig. 4) S&K contrast 'time-dependent lengthening (of G1)' with their model of 'critical lengthening'. Although I am partial to time-dependent lengthening, I appreciate the authors careful comparison. However, several times they declare that critical lengthening has the advantage of 'rapid responses to perturbations and rejection of mis-sensing mutations.' As noted below, case (I) has rapid responses and case (II) has mutant rejection, but neither case has both desirable properties. So the discussion here seems to be misleading. Also, critical lengthening requires somewhat fine-tuning of C close to C_{crit} , which seems (to me) difficult to achieve biochemically. S&K refer to two examples of fine tuning from their own work [58, 59]. I would appreciate a little more explanation of how they think fine tuning might be achieved in the context of cell cycle control.

Minor points

1. In Eq. (1) is the noise term proportional to r or \sqrt{r} ? Maybe it doesn't matter.
2. "In the context of cell cycle regulation, G1 corresponds to negative x , while the G1-S transition occurs at positive x ." I think this sentence is a little too casual. The G1 state is the steady state $x = -$ for $\mu < 0$, and the G1-S transition corresponds to $x(t) \rightarrow +\infty$, either because fluctuations (when $\mu < 0$) drive $x > x_+$ or because the control parameter C exceeds C_{crit} (i.e., $\mu > 0$). "The length of G1...is captured by the typical time taken for x to change sign from negative to positive." Again, too casual for such a crucial feature of the model. Please discuss more clearly the role of fluctuations in driving the control system out of the G1 state and into self-renewal.
3. "In contrast, the RD topology is fragile to invasion of mutants..." Isn't this a fatal flaw of the RD topology, case (I)? The DR topology, case (II), on the other hand, is more resistant to cheater mutations, but it is very slow to recover from low population densities (Fig. 2D). This seems like a pretty serious defect of the DR case. If I understand correctly, both topologies have appealing strengths and significant weaknesses; but this assessment doesn't come through in the Discussion.
4. On p 10, there is a long, speculative application of the model to beta cell dynamics. I am not competent to assess its validity, but the editors should get an expert's opinion on this paragraph.
5. In Fig. 3 legend, "f(N) was replaced by f(α S)..." What is S?
6. Page 10, line 207: "shorten the G1 phase" should be "lengthen the G1 phase," I think.

Reviewer #2:

In their manuscript "Cell cycle criticality as a mechanism for robust cell population control", the authors propose a mechanistic framework for tissue homeostasis when cell fate and cell cycle length are interdependent. They address how - by self-tuning close to a critical point - the cell cycle regulatory network shows several functional benefits ranging from robust set-point control to robustness against the invasion of mutants. To this end, they consider a minimal stochastic model for cell-cycle progression, whose control parameter is itself regulated by the undifferentiated cell population via implicit fast feedback. They argue that tuning to the critical point occurs for two distinct topologies which implement bang-bang control (i.e., reach the set point in minimal time) and efficient mutant rejection, respectively.

In my opinion, the paper addresses an important question in the context of tissue homeostasis and robustness of collective behavior more generally. Unfortunately, the manuscript is not always clearly written and I'm not sure that I could reproduce everything given the information provided. Before recommending publication, I therefore suggest the following changes.

- Eq. (4) is very implicit. Could the authors provide an example for an explicit feedback mechanism that implements Eq. (4)? What is the role of timescales (of the feedback vs. population dynamics vs. Rb phosphorylation)?
- How would noise in the population dynamics and the feedback affect robustness?
- Eq. (3) is a bit mysterious at first. What is the meaning of T here? What about the two different populations? And what about theta? This needs to be more clearly explained. Furthermore, I was very confused at first whether large T is supposed to increase or decrease the probability to differentiate. This is explained later but I found it hard to digest this equation at this point.
- Line 160: "every finite T_S occurs at the vicinity of $C \sim C_{crit}$ ". This statement might need a bit more explanation. For instance, comparing to Fig. 1D, what about values $T_S < 5$?
- Generally, for the coupling between cell-cycle length and cell fate, is the idea that the instantaneous probability for

- differentiation vs renewal changes over time (as maybe suggested by the paragraph starting in line 163), or that external signal simultaneously changes cell cycle length and differentiation probability?
- It is argued that bang-bang control occurs in the RD topology and the set point is achieved in minimal time when starting at low N. But wouldn't this be more true for the DR topology when the initial population size is large?
 - Fig. 2: How was θ chosen here?
 - Line 206/207: Shouldn't the clones extend the G1 phase duration?
 - In the paragraph on the DR topology and its robustness to mutations, it might be worth to explicitly state why the critical state helps (cell cycle length increases by a lot)
 - Figure 3: What is S in $f(\alpha_i S)$?

Reviewer #3:

The authors present a mathematical framework in which an idea is developed for how homeostatically-proliferating tissues might avoid providing a selective advantage to mutant cells that under-sense negative feedback signals. One of the authors had previously published a model in which biphasic responses to feedback signals achieved the same end, and now proposes a different model based, in part, on criticality in the G1-S transition.

A central result of the work, summarized in Fig. 3 and 4, is that a feedback structure based on dividing cells signaling to shorten their own G1 phases, coupled with a shorter G1 enhancing differentiation, results not only in a feedback circuit in which stem cells maintain a constant number, but also one in which mutant cells that are less sensitive to negative feedback are very slow to invade. Moreover, if the relationship between signaling and G1 length is an ultrasensitive one—due to the critical nature of the transition—then mutant cells can become especially disadvantaged.

My major concern about the work is that the connection to real biology is tenuous and quite speculative, even for a theory paper. Yes, the G1-S transition probably does involve a bifurcation, but the evidence that growth-controlling feedback signals act by controlling a G1-S bifurcation parameter is speculative, based mainly, it would appear, on one study of ES cells in culture, cells for which negative feedback-driven homeostasis has not been observed (and probably doesn't occur). Actually, the observation that dividing cells typically control their own numbers in vivo through negative feedback on renewal is itself rather speculative, whereas the alternative that dividing cells are controlled by their differentiated offspring has both experimental and theoretical support (which the authors, incidentally, fail to cite). Finally, the idea that mutant cells fail to overtake normal populations because mutant cells are made to divide more slowly lacks any experimental support that this reviewer is aware of. What little we do know about how mutant cells are prevented from overtaking tissues suggests that active processes involving cell recognition and cell killing are most often involved. In the end we are left with a clever theory, but not a lot of motivation for accepting it.

In addition, several components of the paper that are presented as new theoretical results, while convincing, are either not that relevant or not that much of an advance over previous work. For example, the lengthy discussion about Rb and cyclins in the first half of the paper is interesting, but not particularly relevant to the rest of the paper (i.e. the details of the circuit don't particularly matter to the remainder of the paper, other than it is characterized by bistability). In addition, the general idea that "self-organized criticality" should characterize biological systems became very popular in physics twenty years ago but hasn't achieved much traction in the intervening time. It's not clear that this work adds a tremendous amount more to that discussion. Finally, it is a shame that the manuscript is written in a manner that makes it relatively inaccessible to most biologists. Phenomena that are described in the manuscript tend to be derived from first principles analyses of equations, even when the phenomena have other examples in biology that could be used to illustrate them. For example, critical slowing when systems cross bifurcations is a generic phenomenon, and appears in other areas of biology, such as ecosystem science, where helpful analogies could have been drawn. Greater use of analogies and simulations, such as those that appear in Fig. 2C and D, could have also made the early results of the manuscript more accessible, and could have spared biologists from having to wade through stochastic and partial differential equations that, in the end, are only important in their broadest interpretation. Ultimately, the combination of being highly speculative while also being difficult for experimentalists to wade through is likely to diminish the potential impact of this work, which does contain some interesting ideas that are potentially worth sharing.

Reviewer #4:

In this manuscript, the authors develop a theoretical framework to analyze how coupling between cell cycle progression (specifically G1 lengthening) and cell fate decisions enables robust tissue population control by stem cells. By modeling the G1-S transition as a noisy saddle-node bifurcation—where mitogens acting on cell cycle proteins, such as cyclin D and cyclin E, act as control parameters for the G1-S transition—the authors demonstrate that competitive feedback self-tunes the system to a critical bifurcation point where dynamics slow down. This critical positioning confers two key advantages: (1) robust population (of stem cells) set-point maintenance and (2) efficient rejection of mis-sensing mutants. The authors test their model predictions using experimental data from human embryonic stem cells (hESCs) under varying Wnt signaling conditions including non-linear G1 lengthening and amplified variability near the critical point.

Overall, the authors bring to light two interesting and somewhat counterintuitive predictions: (1) positioning of stem cells near the critical point (where cell-cell variability is amplified) for G1-S transition can confer robust population control, and (2) there might be an inherent trade-off between the ability of stem cells to rapidly regenerate a tissue and avoiding mutant takeover. However,

there are a few major concerns that need to be addressed to significantly strengthen the manuscript as highlighted below.

Major concerns:

1. Robustness of results to choice of parameter values is unclear: Manuscript code and detailed tabulation of parameter values used for each figure are missing. For example, parameter values for V , reasoning for a choice of 2 for the Hill coefficient for the cooperative Rb phosphorylation via the E2F/CycE cascade ($\gamma P \propto 1/(1+(r/k_2)^2)$), and initial conditions for simulations are not mentioned. This is critical for both reproducing the authors' results and for testing how the choice of parameter values impacts the results. For this purpose, the authors should tabulate the parameter values used for each figure (maybe in the figure captions or the supplementary section) and comment about the range of parameter values for which their results hold.
2. Non-linear G1 lengthening in hESCs upon Wnt removal is not sufficiently supported: In Fig. 1D, the sharp non-linear increase in G1 length of hESCs when Wnt activity/mitogen signaling is reduced needs to be better supported, especially because the accuracy of fitting for the authors' model is not quantified. One line of evidence to better support this claim would be to (1) quantify the strength of fitting for the authors' model (say R^2 value), and (2) perform a linear fit and compare the R^2 values for the linear model to the authors' model.
3. Self/fine-tuning of the system near the critical point and corresponding population control for RD and DR topologies should be characterized for smaller stem cell niches: The authors highlight that cell-cell variability is amplified near the critical point (Fig. 1E) and fine-tuning near the critical point is necessary for population control (as finite T_s values occur close to C_{crit}). It is somewhat counterintuitive that robust population control can only be achieved at the same point where cell-cell variability is also enhanced. A natural question is what happens if the stem cell niche is much smaller (say ~ 10 cells)? Would the increased cell-cell variability near the critical point prevent population control for smaller stem cell niches?

Minor concerns:

1. Fig. 1C is unclear -> the "delay" black line should be explained in more detail to contextualize the stretched exponential fit that predicts G1 lengthening in the vicinity of C_{crit}
2. Legend/colors in the population size plot in Fig. 3C seems to be flipped
3. It should be made explicit in the Introduction that the paper focusses on stem cell self-renewal vs differentiation as opposed to population control of different types of differentiated cells. In line with this, first sentence of the abstract could be changed to "tissue homeostasis requires a precise balance between stem cell self-renewal and differentiation".

Dear Dr. Hou,

Thank you for the opportunity to revise our manuscript, "Cell cycle criticality as a mechanism for robust cell population control" (MSB-2025-13167). We are grateful to you and the four reviewers for the thoughtful and constructive feedback, which has enabled us to substantially improve the paper.

We have undertaken a major revision to address all the points raised. A key focus was to further strengthen the biological grounding of our work and improve its accessibility, especially for the non-mathematical readership.

The most significant changes include:

1. **Strengthening biological connections:** We have added a new section and a new figure (Figure 2) that directly connects a core prediction of our model – the emergence of a prolonged, reversible "intermediate state of the cell cycle" due to critical slowing down – to recent, high-impact experimental findings on the Rb-E2F switch (Konagaya et al., *Nature*, 2024). This provides timely evidence for the validity and relevance of our proposed mechanism.
2. **Improving accessibility and generality:** To make the paper more accessible to a broad audience, we have added further conceptual illustrative explanations throughout the text. We have also expanded the main text and the Appendix to include concrete examples of biological circuits that would implement our proposed RD and DR topologies, and we reinforce our original discussion to formally demonstrate that our conclusions hold even when feedback is mediated by a differentiated cell population.
3. **Enhancing reproducibility and rigor:** We have added a comprehensive new appendix detailing all simulation parameters and methodologies, and we have performed quantitative tests as requested by the reviewers. Furthermore, we have made all aspects of the simulation code publicly available.

We have also addressed all other minor points and editorial requirements. A detailed point-by-point response to each reviewer comment is included in the attached letter.

We believe these revisions substantially improve the clarity and impact of the manuscript.

Sincerely,

Omer Karin and Benjamin D. Simons

9th Jul 2025

Manuscript Number: MSB-2025-13167

Title: Cell cycle criticality as a mechanism for robust cell population control

Author: Benjamin Simons

Omer Karin

Dear Dr Karin,

Thank you for submitting your manuscript to Molecular Systems Biology. First, I would like to apologize for the delay in the review process, which was due to the late arrival of the reviewers' reports. We have now received feedback from all four reviewers who agreed to evaluate your work. As you will see from their comments below, the reviewers recognize the interest and potential significance of your study. However, they have also raised a number of substantial concerns that will need to be addressed through a major revision.

In particular, Reviewers #2 and #3 have expressed concerns regarding several assumptions used in your model. Additionally, Reviewer #3 questioned the overall relevance of the work to real biological systems. These are critical issues that require careful consideration, and we encourage you to strengthen the biological grounding of your study accordingly.

Please note that due to formatting issues with mathematical equations in the online system, Reviewer #1 has submitted their comments as a separate Word document, which is attached for your reference.

All other reviewer comments should be addressed comprehensively as well. As per our editorial policy, we generally allow only one round of major revision, so it is important that your responses be as thorough and complete as possible.

On a more editorial level, we would ask you to address the following issues:

- Please provide a .docx formatted version of the manuscript text (including legends for main figures, EV figures and tables). Please make sure that the changes are highlighted to be clearly visible.

- Please provide individual production quality figure files as .eps, .tif, .jpg (one file per figure).

- Please provide a .docx formatted letter INCLUDING the reviewers' reports and your detailed point-by-point responses to their comments. As part of the EMBO Press transparent editorial process, the point-by-point response is part of the Review Process File (RPF), which will be published alongside your paper.

-Please note that all corresponding authors are required to supply an ORCID ID for their name upon submission of a revised manuscript.

-We replaced Supplementary Information with Expanded View (EV) Figures and Tables that are collapsible/expandable online (see examples in <http://msb.embopress.org/content/11/6/812>). A maximum of 5 EV Figures can be typeset. EV Figures should be cited as 'Figure EV1, Figure EV2" etc... in the text and their respective legends should be included in the main text after the legends of regular figures.

Additional Tables/Datasets should be labeled and referred to as Table EV1, Dataset EV1, etc. Legends have to be provided in a separate tab in case of .xls files. Alternatively, the legend can be supplied as a separate text file (README) and zipped together with the Table/Dataset file. For the figures and tables that you do NOT wish to display as Expanded View figures, they should be bundled together with their legends in a single PDF file called *Appendix*, which should start with a short Table of Content. Each legend should be below the corresponding Figure/Table in the Appendix. Appendix figures and tables should be referred to in the main text as: "Appendix Figure S1, Appendix Figure S2, Appendix Table S1" etc. See detailed instructions regarding expanded view here:

<https://www.embopress.org/page/journal/17444292/authorguide#expandedview>.

-Before submitting your revision, primary datasets (and computer code, where appropriate) produced in this study need to be deposited in an appropriate public database

(see <http://msb.embopress.org/authorguide> - dataavailability

<https://www.embopress.org/page/journal/17444292/authorguide#dataavailability>).

The accession numbers and database should be listed in a formal "Data Availability" section (placed after Materials & Method) that follows the model below (see also

<https://www.embopress.org/page/journal/17444292/authorguide#dataavailability>). Please note

that the Data Availability Section is restricted to new primary data that are part of this study.

Data availability

- RNA-Seq data: Gene Expression Omnibus GSE46843

(<https://www.ncbi.nlm.nih.gov/geo/query/acc.cgi?acc=GSE46843>)

- [data type]: [name of the resource] [accession number/identifier/doi] ([URL or identifiers.org/DATABASE:ACCESSION])

Additional information on source data and instruction on how to label the files are available

<https://www.embopress.org/page/journal/14693178/authorguide#sourcedata>

- Our journal encourages inclusion of *data citations in the reference list* to directly cite datasets that were re-used and obtained from public databases. Data citations in the article text are distinct from normal bibliographical citations and should directly link to the database records from which the data can be accessed. In the main text, data citations are formatted as follows: "Data ref: Smith et al, 2001". In the Reference list, data citations must be labeled with "[DATASET]". A data reference must provide the database name, accession number/identifiers and a resolvable link to the landing page from which the data can be accessed at the end of the reference. Further instructions are available at <https://www.embopress.org/page/journal/17444292/authorguide#referencesformat> >.

- We updated our journal's competing interests policy in January 2022 and request authors to consider both actual and perceived competing interests. Please review the policy <https://www.embopress.org/competing-interests> and update your competing interests if necessary. Please use the heading "Disclosure statement and competing interests".

- All Materials and Methods need to be described in the main text using our 'Structured Methods' format. According to this format, the Methods section includes a Reagents and Tools Table (listing key reagents, experimental models, software and relevant equipment and including their sources and relevant identifiers) followed by a Methods and Protocols section describing the methods, ideally using a step-by-step protocol format. The aim is to facilitate adoption of the methodologies across labs.

Please download and fill our Reagents and Tools Table template (.docx), which you can find in our author guidelines:

<https://www.embopress.org/page/journal/17444292/authorguide#structuredmethods>.

-Regarding data quantification:

Please ensure to specify the name of the statistical test used to generate error bars and P values, the number (n) of independent experiments (please specify technical or biological replicates) underlying each data point and the test used to calculate p-values in each figure legend. Discussion of statistical methodology can be reported in the materials and methods section, but figure legends should contain a basic description of n, P and the test applied. Graphs must include a description of the bars and the error bars (s.d., s.e.m.). Please also include scale bars in all microscopy images.

- Please provide a "standfirst text" summarizing the study in one or two sentences (approximately 250 characters, including space), three to four "bullet points" highlighting the main findings and a "synopsis image" (550px width and 400-600 px height, PNG format) to highlight the paper on our homepage.

Here are a couple of examples:

<https://www.embopress.org/doi/10.15252/msb.20199356>

<https://www.embopress.org/doi/10.15252/msb.20209475>

<https://www.embopress.org/doi/10.15252/msb.209495>

When you resubmit your manuscript, please download our CHECKLIST

(<https://www.embopress.org/pb-assets/embo-site/EMBO%20Press%20Author%20Checklist-1642513524327.xlsx>) and include the completed form in your submission.

Please note that the Author Checklist will be published alongside the paper as part of the transparent process

(<https://www.embopress.org/page/journal/17444292/authorguide#transparentprocess>).

If you feel you can satisfactorily deal with these points and those listed by the referees, you may wish to submit a revised version of your manuscript. Please attach a covering letter giving details of the way in which you have handled each of the points raised by the referees. A revised manuscript will be once again subject to review and you probably understand that we can give you no guarantee at this stage that the eventual outcome will be favorable.

I look forward to receiving your manuscript soon.

Kind regards,
Jingyi

Jingyi Hou, PhD
Senior Editor
Molecular Systems Biology

We realize that it is difficult to revise to a specific deadline. In the interest of protecting the conceptual advance provided by the work, we recommend a revision within 3 months (27th Oct 2025). Please discuss the revision progress ahead of this time with the editor if you require more time to complete the revisions. Use the link below to submit your revision:

*** PLEASE NOTE *** As part of the EMBO Press transparent editorial process initiative (see our Editorial at <https://dx.doi.org/10.1038/msb.2010.72>), Molecular Systems Biology publishes online a Review Process File with each accepted manuscripts. This file will be published in conjunction with your paper and will include the anonymous referee reports, your point-by-point response and all pertinent correspondence relating to the manuscript. If you do NOT want this File to be published, please inform the editorial office at contact@molsystbiol.org within 14 days upon receipt of the present letter.

Reviewer #1:

I rated the 'validity of conclusions' medium because of some major revisions requested below.

My report copied below is difficult to read because all the mathematical notation was screwed up. I sent a copy of the Word document to the MSB email address, but I don't know if it got attached to this review.

Summary

The authors, Simons & Karin (S&K), put forward a mathematical model to understand how proliferating stem cells maintain an appropriate balance between self-renewal and differentiation, and, at the same time, eliminate deleterious mutants that outgrow wild-type cells. Although their model is quite abstract, it is based soundly on the experimentally established fact that the G1/S transition in mammalian cells is controlled by a saddle-node (SN) bifurcation. Their basic idea is that cell differentiation occurs from G1 phase of the cell cycle; however, as a control parameter C (e.g., the activity of CycD-dependent kinase) increases past a critical point, C_{crit} , the stable G1 state is eliminated at a SN bifurcation, and the cell makes an irreversible commitment to S-G2-M phase and cell division (i.e., self-renewal). From a mathematical point-of-view, all SN bifurcations 'look the same' and can be described by a simple stochastic differential equation

$$dx/dt = x^2 + \mu + \text{noise}$$

where x denotes the state of the cell cycle and $\mu = C - C_{crit}$ is the 'distance' of the control system from the SN bifurcation. Note: for $\mu = C - C_{crit} < 0$, $x(t)$ has two steady-state solutions, $x_- = -\sqrt{\mu}$ and $x_+ = +\sqrt{\mu}$, where $x_- < 0$ corresponds to the stable G1 state, and $x_+ > 0$ is an unstable saddle point. As μ passes through 0, the two steady states coalesce and disappear, and for $\mu > 0$, $x(t) \rightarrow +\infty$, which the authors interpret as an irreversible commitment to self-renewal. A generic property of SN bifurcations in the presence of noise, which is crucial to S&K's theory, is that, for C close to but less than C_{crit} , i.e., for $|\mu| \approx 0$, there is a finite time T when fluctuations may drive the control system off the G1 state (x_-), past the saddle point (x_+), and into DNA synthesis and division. Furthermore, transition time gets longer as the distance from criticality increases: $\log T \sim |\mu|^{3/2}$. Notice that $T = \Phi(C)$, with $\Phi(C_{crit}) = 0$ and $\partial\Phi/\partial C < 0$. The authors associate this time T to the duration of G1 phase and, hence, to the cell generation time (assuming the duration of S-G2-M is fixed and small, relative to T). In this case, the number $N(t)$ of undifferentiated stem cells is given by

$$dN/dt = N/T \theta(T)$$

where $\theta(T)$ specifies how the net rate of self-renewal depends on G1 duration. (Note: $\theta(T)$ can be < 0 because there are always some stem cells lost by cell death.) The tissue achieves

balanced self-renewal ($N(t) = NS = \text{constant}$, $dN/dt = 0$) for $\theta(TS) = 0$ at some $T = TS$. Now S&K point out that, since all finite T 's are found in the vicinity of $C = C_{\text{crit}}$, the steady-state population density is found in the vicinity of C_{crit} , i.e., $NS = f^{-1}(C_{\text{crit}})$, where $C = f(N)$ denotes the dependence of the control parameter C on stem-cell density N . The authors point out that the fixed point of self-renewal, NS , is stable if and only if the sign of $[\partial\theta/\partial T]$ is the same as the sign of $[\partial f/\partial N]$. Clearly there are two cases: (I) both signs are negative or (II) both signs are positive. The authors go on to study these two cases in detail, showing that in case (I) self-renewal occurs rapidly, whereas case (II) is more resistant to being displaced by mutants that are biased toward self-renewal ('cheater' mutants).

General remarks

This manuscript presents a very elegant and (to my knowledge) novel approach to the question of tissue maintenance by self-renewing populations of stem cells, with particular emphasis on the important question of stability with respect to cheater mutations. The paper is well motivated and, for the most part, very clear. The figures are attractive and informative. The advances are primarily conceptual. The paper will be of interest to biological physicists, mathematical biologists, and cell biologists with a sophisticated understanding of dynamical systems theory (saddle-node bifurcations, stochastic differential equations, barrier crossing). The paper is very suitable for publication in *Molecular Systems Biology*.

We thank the referee for this positive and encouraging endorsement of the work.

Major points

1. "The authors point out that the fixed point of self-renewal, NS , is stable if and only if the sign of $[\partial\theta/\partial T]$ is the same as the sign of $[\partial f/\partial N]$." I am sure the authors are correct in this statement, but it is not obvious from their Eqs. (3-5). It took me some time to convince myself of the claim, and I had to introduce the function " $T = \Phi(C)$, with $\Phi(C_{\text{crit}}) = 0$ and $\partial\Phi/\partial C < 0$ " in order to connect the function $f(N)$ to $\theta(T)$. Because Eq. (6) is central to the entire argument, the authors need to provide a convincing demonstration of this claim.

We addressed this by adding a stability analysis section to the Supplementary Information ("Stability of the Population Set-Point") as detailed below:

2 Stability of the Population Set-Point

To determine the stability of the homeostatic population set-point, N_s , we perform a linear stability analysis. The system dynamics are described by the population growth rate, which depends on the G1 duration, T ,

$$\frac{\dot{N}}{N} = \frac{1}{T}\theta(T), \quad (3)$$

and a feedback loop where the G1 duration is controlled by the population size, N . The function $\theta(T)$ captures the net self-renewal bias and we assume that it has a single intersection point with the T axis (and therefore a single fixed point), at least within the physiologically relevant range of T . This feedback is mediated by the control parameter, C , such that $C = f(N)$ and $T = \Phi(C)$, where $\Phi(C)$ is the function that maps the control parameter to the G1 duration. A steady-state is achieved at a population size N_s where $\dot{N} = 0$, which requires that $\theta(T_s) = 0$ for some steady-state G1 duration T_s .

Consider a small perturbation, δN , from the steady-state, such that $N = N_s + \delta N$. This perturbation propagates through the feedback loop:

1. The change in population size alters the control parameter C :

$$\delta C \approx \left. \frac{df}{dN} \right|_{N_s} \delta N$$

2. The change in C alters the G1 duration T :

$$\delta T \approx \left. \frac{d\Phi}{dC} \right|_{C_s} \delta C$$

3. The change in T alters the net growth bias $\theta(T)$:

$$\delta\theta \approx \left. \frac{d\theta}{dT} \right|_{T_s} \delta T$$

Combining these steps, the response of the growth bias to the initial population perturbation is:

$$\delta\theta \approx \left(\left. \frac{d\theta}{dT} \right|_{T_s} \left. \frac{d\Phi}{dC} \right|_{C_s} \left. \frac{df}{dN} \right|_{N_s} \right) \delta N \quad (4)$$

For the set-point N_s to be stable, a positive perturbation ($\delta N > 0$) must induce a negative growth rate ($\dot{N} < 0$, which implies $\theta < 0$ and thus $\delta\theta < 0$). This requires the term in the parenthesis to be negative:

$$\left. \frac{d\theta}{dT} \right|_{T_s} \left. \frac{d\Phi}{dC} \right|_{C_s} \left. \frac{df}{dN} \right|_{N_s} < 0 \quad (5)$$

2

From the fundamental properties of the saddle-node bifurcation (Fig. 1C), an increase in the control parameter C (e.g., mitogen level) shortens the G1 duration T . Therefore, the derivative $\frac{d\Phi}{dC}$ is negative. For the entire expression to be negative, the product of the remaining two derivatives must be positive:

$$\left. \frac{d\theta}{dT} \right|_{T_s} \left. \frac{df}{dN} \right|_{N_s} > 0 \quad (6)$$

This condition is only satisfied if both derivatives have the same sign, leading to the stability criterion presented in the main text:

$$\text{sign} \left(\frac{d\theta}{dT} \right) = \text{sign} \left(\frac{df}{dN} \right) \quad (7)$$

2. In the Discussion (Fig. 4) S&K contrast 'time-dependent lengthening (of G1)' with their model of 'critical lengthening'. Although I am partial to time-dependent lengthening, I appreciate the authors careful comparison. However, several times they declare that critical lengthening has

the advantage of 'rapid responses to perturbations and rejection of mis-sensing mutations.' As noted below, case (I) has rapid responses and case (II) has mutant rejection, but neither case has both desirable properties.

We agree, and we have revised the text throughout to make this distinction clearer and avoid any ambiguity.

So the discussion here seems to be misleading. Also, critical lengthening requires somewhat fine-tuning of C close to C_{crit} , which seems (to me) difficult to achieve biochemically. S&K refer to two examples of fine tuning from their own work [58, 59]. I would appreciate a little more explanation of how they think fine tuning might be achieved in the context of cell cycle control.

The reviewer correctly notes that externally fine-tuning a system to a critical point is difficult. However, a key strength of our model is that, as an emergent property of the negative feedback loop, the system can self-tune to this point. Any stable, extended G1 phase can only exist near the bifurcation point (Fig. 1C). Therefore, for the population to reach a homeostatic set-point based on a prolonged G1, the feedback must dynamically drive and maintain the system in this critical regime, as we demonstrate in the current Figure 3. The fine-tuning is not a fragile assumption but a robust outcome of the closed-loop dynamics. We have now made this central point clearer in our revised discussion and we welcome the reviewer's request for clarification.

Minor points

1. In Eq. (1) is the noise term proportional to r or \sqrt{r} ? Maybe it doesn't matter.

While in our model we chose to represent fluctuations with a noise term proportional to r , we agree with the reviewer that the specific form of this term does not affect our conclusions. This is because the key dynamics occur near the bifurcation's critical point, where the system state r is confined to a narrow range around its critical value, r_{crit} . In this regime, the state-dependent noise effectively behaves like a noise source with a nearly constant amplitude, and the universal properties of the saddle-node bifurcation hold. We have now clarified this in the revised manuscript.

2. "In the context of cell cycle regulation, G1 corresponds to negative x , while the G1-S transition occurs at positive x ." I think this sentence is a little too casual. The G1 state is the steady state $x^- < 0$ for $\mu < 0$, and the G1-S transition corresponds to $x(t) \rightarrow +\infty$, either because fluctuations (when $\mu < 0$) drive $x > x^+$ or because the control parameter C exceeds C_{crit} (i.e., $\mu > 0$). "The length of G1...is captured by the typical time taken for x to change sign from negative to positive." Again, too casual for such a crucial feature of the model. Please discuss more clearly the role of fluctuations in driving the control system out of the G1 state and into self-renewal.

We have revised the paragraph to more precisely define the G1-S transition, including distinguishing between noise-driven escape in the sub-critical regime and the "ghost" effect of critical slowing down around the critical point:

The saddle-node bifurcation has three key properties that will play an important role in our analysis. The first is that in the vicinity of $C \approx C_{\text{crit}}$, the dynamics of the system are captured by a one-dimensional noisy process,

$$dx = (x^2 + \mu) dt + \xi dW, \quad (2)$$

where x denotes the rescaled coordinates that parametrize the bistable switch, ξ corresponds to noise amplitude by the critical point, time t is measured in rescaled coordinates, and $\mu \propto C - C_{\text{crit}}$ is the (rescaled) distance from the bifurcation. The G1 state is associated with negative values of x , while the G1-S transition corresponds to the irreversible progression of x to positive values. The length of G1, denoted T , is the time taken for this transition, which can be prolonged by distinct mechanisms depending on the sign of μ . For $\mu < 0$ (sub-critical), the system resides in a stable G1 state, and the transition is a stochastic event requiring a noise-driven escape over an unstable barrier; here, T corresponds to the mean first passage time. Conversely, for $\mu \geq 0$ (super-critical), the G1 state is no longer stable, but, as long as μ is small, the system's deterministic progression may be significantly delayed as it passes through a "ghost" of the former fixed points, a phenomenon known as critical slowing down (Koch, Nandan, Ramesan, Tyukin, et al. 2024; Strogatz 2024). Otherwise, for large μ , G1 progression is rapid.

3. "In contrast, the RD topology is fragile to invasion of mutants..." Isn't this a fatal flaw of the RD topology, case (I)? The DR topology, case (II), on the other hand, is more resistant to cheater mutations, but it is very to recover from low population densities (Fig. 2D). This seems like a pretty serious defect of the DR case. If I understand correctly, both topologies have appealing strengths and significant weaknesses; but this assessment doesn't come through in the Discussion.

We have now edited the discussion to include a broad summary of the potential limitations of the DR and RD topologies and how complementary mechanisms can help overcome them:

Our model reveals a fundamental trade-off between the RD and DR topologies. The RD topology, while enabling rapid population recovery from low cell numbers, is inherently fragile to invasion by "cheater" mutants with shorter G1 phases. Conversely, the DR topology is robust against such mutations, as it preferentially removes faster-cycling cells through differentiation before they can self-renew. However, this robustness comes at a significant cost: a severely limited ability to recover from population bottlenecks.

These inherent fragilities may be overcome by complementary mechanisms. For the RD topology, for instance, biphasic regulation, where growth becomes negative at very fast division rates, could prevent mutant accumulation, albeit at the risk of population collapse (Karin and Alon 2017). For the DR topology, recovery from low population abundances could be boosted by periodic signals that force cells into the cycle. Such a mechanism is observed in spermatogenesis, where the seminiferous cycle provides a periodic signal that drives stem cell division approximately every 10 days. Following division, these stem cells appear to enter a transient state that is more susceptible to differentiation but can be rescued by self-renewal factors present at that stage of the cycle (Nakagawa et al. 2021). This process may potentially act as a variation of the DR topology: it prevents runaway "cheater" mutants that escape the seminiferous cycle's control, while the periodic nature of the signal ensures the population can recover from large perturbations, as has been observed experimentally (Kitadate et al. 2019).

4. On p 10, there is a long, speculative application of the model to beta cell dynamics. I am not competent to assess its validity, but the editors should get an expert's opinion on this paragraph.
5. In Fig. 3 legend, "f(N) was replaced by f(α iS)..." What is S?
6. Page 10, line 207: "shorten the G1 phase" should be "lengthen the G1 phase," I think.

We thank the reviewer for identifying these issues, which are now fixed in the revised manuscript.

Reviewer #2:

In their manuscript "Cell cycle criticality as a mechanism for robust cell population control", the authors propose a mechanistic framework for tissue homeostasis when cell fate and cell cycle length are interdependent. They address how - by self-tuning close to a critical point - the cell cycle regulatory network shows several functional benefits ranging from robust set-point control to robustness against the invasion of mutants. To this end, they consider a minimal stochastic model for cell-cycle progression, whose control parameter is itself regulated by the undifferentiated cell population via implicit fast feedback. They argue that tuning to the critical point occurs for two distinct topologies which implement bang-bang control (i.e., reach the set point in minimal time) and efficient mutant rejection, respectively.

In my opinion, the paper addresses an important question in the context of tissue homeostasis and robustness of collective behavior more generally. Unfortunately, the manuscript is not always clearly written and I'm not sure that I could reproduce everything given the information provided. Before recommending publication, I therefore suggest the following changes.

We thank the referee for recognizing the importance of our work and for their constructive recommendations, which we address below.

- Eq. (4) is very implicit. Could the authors provide an example for an explicit feedback mechanism that implements Eq. (4)? What is the role of timescales (of the feedback vs. population dynamics vs. Rb phosphorylation)?

We now address this in a new Supplementary Information section ("Mechanistic Examples of Population Feedback"), where we provide explicit mechanistic models for population level feedback and discuss the role of timescales:

B Mechanistic Examples of Population Feedback

To provide a more concrete biological grounding for the feedback function $C = f(N)$, we consider a simplified model based on a diffusible ligand that illustrates how both the Renewal precedes Differentiation (RD) and Differentiation precedes Renewal (DR) topologies can arise.

Consider a ligand, l , that is secreted at a constant rate, p , by the surrounding niche and is consumed or degraded by the stem cells, N , at a per-capita rate k_{deg} . The dynamics of the ligand concentration can be described as:

$$\frac{dl}{dt} = p - k_{deg}Nl \quad (8)$$

Assuming the ligand dynamics are fast relative to cell division, the concentration reaches a steady-state ($dl/dt = 0$) given by $l_{ss} = p/(k_{deg}N)$. The resulting feedback topology now depends entirely on whether the ligand acts as a mitogen or an inhibitor.

1 Case 1: Ligand is a Mitogen (RD Topology)

If the ligand promotes cell cycle progression, the control parameter C will be proportional to the ligand concentration, $C = \alpha l_{ss}$. The feedback function is therefore:

$$C = f(N) = \frac{\alpha p}{k_{deg}N} \quad (9)$$

The derivative of this function, df/dN , is negative. Stability requires that $d\theta/dT < 0$, meaning a longer G1 phase is associated with differentiation. This corresponds to the RD topology.

2 Case 2: Ligand is an Inhibitor (DR Topology)

If the ligand inhibits cell cycle progression (e.g., represses Cyclin D), the control parameter C will be inversely proportional to the ligand concentration, $C = \beta/l_{ss}$. The feedback function becomes:

$$C = f(N) = \frac{\beta k_{deg}N}{p} \quad (10)$$

The derivative of this function, df/dN , is positive. Stability now requires that $d\theta/dT > 0$, meaning a longer G1 phase is associated with self-renewal. This corresponds to the DR topology.

3 The Role of Timescales and the Quasi-Steady-State Assumption

Our model relies on a separation of timescales. The intracellular dynamics of the Rb/E2F network operate on a timescale of minutes to hours. In contrast, population dynamics, characterized by cell division and changes in the total cell number N , occur on a much slower timescale of many hours to days.

The environmental feedback, mediated by the ligand l , must operate on a timescale that is fast relative to the timescale of cell division. This is the justification for the quasi-steady-state assumption ($dl/dt \approx 0$) used above. If the ligand dynamics were slower than cell division, a cell would be responding to an outdated environmental signal that reflects a past population size. Such a significant delay would introduce instability into the feedback loop, potentially leading to large oscillations around the homeostatic set-point rather than stable regulation.

- How would noise in the population dynamics and the feedback affect robustness?

We thank the reviewer for this important question regarding the system's robustness to noise. Our model is indeed robust to both sources of noise mentioned by the reviewer. First, noise in the population dynamics is most prominent in small populations. As we demonstrate in our simulations for small stem cell niches (a point also raised by Reviewer #4), the homeostatic set-point is maintained even with significant demographic fluctuations, although the probability of stochastic extinction naturally increases, which is a realistic feature of small systems.

Second, noise in the feedback is already an intrinsic feature of our simulations. The feedback signal is a function of the number of stem cells, which is a stochastically fluctuating integer. This inherently introduces noise into the control parameter C at every time step. Theoretically, this

noise in the feedback translates directly to fluctuations in the bifurcation parameter, μ . This has a similar effect to the intrinsic noise (ξ): both contribute to the probability of a cell crossing the G1/S transition barrier, thus desynchronizing the population. Perhaps counter-intuitively, in our model, fluctuations actually increase robustness as they prevent uniform population dynamics (all cells dividing or differentiating at the same time).

- Eq. (3) is a bit mysterious at first. What is the meaning of T here? What about the two different populations? And what about θ ? This needs to be more clearly explained. Furthermore, I was very confused at first whether large T is supposed to increase or decrease the probability to differentiate. This is explained later but I found it hard to digest this equation at this point.

We apologize if the previous narrative were confusing on the definitions. To address this concern, we have added a paragraph following Eq. 3 to explain more carefully the meaning of the parameter T and the function θ :

$$\frac{\dot{N}}{N} = T^{-1}\theta(T), \quad (3)$$

This equation describes the rate of change for the stem cell population N at a fixed cell division rate. The parameter T corresponds to the G1 duration of a cell, which assumed to dominate the overall cell division time, so the term proportional to $1/T$ converts the outcome of a discrete cell division event into a continuous growth rate. The function $\theta(T)$ captures the net self-renewal bias, which is defined by the probabilities of the different fate outcomes. Denoting the probability of a self-renewing division as $P_R(T)$ and a differentiating division as $P_D(T)$, the bias is given by $\theta(T) = P_R(T) - P_D(T)$. For instance, if a cell division of duration T yields on average the same number of differentiated and stem cells, then $\theta(T) = 0$. Critically, our model does not presuppose a specific relationship between G1 length and cell fate. Instead, as we will show, the nature of this dependence is what defines two distinct and functionally opposing regulatory strategies.

- Line 160: "every finite T_S occurs at the vicinity of $C \sim C_{crit}$ ". This statement might need a bit more explanation. For instance, comparing to Fig. 1D, what about values $T_S < 5$?

This has been corrected in the text to highlight that we consider non-zero and extended values of T_S .

- Generally, for the coupling between cell-cycle length and cell fate, is the idea that the instantaneous probability for differentiation vs renewal changes over time (as maybe suggested by the paragraph starting in line 163), or that external signal simultaneously changes cell cycle length and differentiation probability?

We thank the reviewer for this clarifying question. Our model aligns with the first interpretation: the probability of differentiation effectively changes as a function of G1 duration. In practice, it is likely that external signals may modulate both cell cycle duration and the differentiation probability function. However, we do not expect this to affect our conclusions as long as the fixed point occurs at an extended cell cycle duration.

- It is argued that bang-bang control occurs in the RD topology and the set point is achieved in minimal time when starting at low N . But wouldn't this be more true for the DR topology when the initial population size is large?

We agree with the reviewer that the DR topology provides a rapid response to an overly large population by promoting differentiation through short cell cycles. However, our characterization of the RD topology as providing "bang-bang control" specifically focused on the challenge of reaching a set-point from a low population size, as is critical during development and tissue repair. In this context, the RD topology is superior, as it couples rapid cell cycles with self-renewal. While the RD topology might seem slow to respond to overpopulation due to its long G1 cycles, this is not necessarily a disadvantage. As we now clarify in the revised manuscript, this issue is circumvented if cells commit to differentiation before their long G1 phase is concluded (a typical feature of terminal differentiation). See revised paragraph:

While both the RD and DR topologies exhibit sensitive responses to changes in cell abundance, they have very different dynamics away from the set point. When the number of cells N is much lower than the set point, both topologies can result in pure self-renewal, but the dynamics differ. In the RD topology, self-renewal occurs rapidly due to short cell cycle times. Only as the population approaches its steady state does a slowdown in G1 occur, resulting in large-scale cell cycle arrest (Figure 3C). In control theory, this is known as *bang-bang control*, providing an optimal response for reaching a set point in minimal time (Itzkovitz et al. 2012). This type of dynamic appears to be widespread in developmental settings (Itzkovitz et al. 2012; Kicheva et al. 2014; Lange et al. 2009; Molina, Bonnet, et al. 2022). Conversely, the DR topology exhibits sluggish recovery from low cell numbers due to the extension of division times (Figure 3D). However, when the population is too large, the DR topology can respond rapidly with short cell cycles that promote differentiation. While the RD topology might seem disadvantaged in this scenario due to its longer cell cycles, this can be easily circumvented if cells commit to differentiation before G1 is concluded, as is common in terminal differentiation. We therefore propose that the RD topology provides a particularly effective mechanism for rapid tissue development and growth.

- Fig. 2: How was theta chosen here?

We have now added an explanation about this to the simulation section:

The precise choice of the fate function $\theta(T)$ is not critical for our conclusions, as long as the steady-state balance ($\theta(T_s) = 0$) occurs at a sufficiently large T_s that requires critical lengthening.

- In the paragraph on the DR topology and its robustness to mutations, it might be worth to explicitly state why the critical state helps (cell cycle length increases by a lot)

We have now revised the manuscript to explicitly state that the critical state enhances mutant rejection in the DR topology by amplifying small mutational differences in signal sensitivity into large, non-linear increases in G1 length.

- Line 206/207: Shouldn't the clones extend the G1 phase duration?

- Figure 3: What is S in $f(\alpha_i S)$?

We thank the reviewer for identifying these issues, which are now fixed in the revised manuscript.

Reviewer #3:

The authors present a mathematical framework in which an idea is developed for how homeostatically-proliferating tissues might avoid providing a selective advantage to mutant cells that under-sense negative feedback signals. One of the authors had previously published a model in which biphasic responses to feedback signals achieved the same end, and now proposes a different model based, in part, on criticality in the G1-S transition.

A central result of the work, summarized in Fig. 3 and 4, is that a feedback structure based on dividing cells signaling to shorten their own G1 phases, coupled with a shorter G1 enhancing differentiation, results not only in a feedback circuit in which stem cells maintain a constant number, but also one in which mutant cells that are less sensitive to negative feedback are very slow to invade. Moreover, if the relationship between signaling and G1 length is an ultrasensitive one—due to the critical nature of the transition—then mutant cells can become especially disadvantaged.

My major concern about the work is that the connection to real biology is tenuous and quite speculative, even for a theory paper. Yes, the G1-S transition probably does involve a bifurcation, but the evidence that growth-controlling feedback signals act by controlling a G1-S bifurcation parameter is speculative, based mainly, it would appear, on one study of ES cells in culture, cells for which negative feedback-driven homeostasis has not been observed (and probably doesn't occur).

We thank the reviewer for raising this critical point about the biological grounding of our model. We understand that the concern is twofold: (1) whether growth-controlling signals act by tuning a G1-S bifurcation parameter, and (2) whether this process is part of a tissue-level negative feedback loop.

We believe that, as mentioned by the referee, the assumption that the G1/S transition is controlled by a bifurcation is not particularly controversial and supported directly by experiment, such as Yao et al. 2008, Cappell et al. 2016, and Schwarz et al. 2018 (cited in the manuscript), and the general phenomenology of an irreversible transition at high E2F activity. These studies and others typically point to the regulation of cell cycle length by factors such as Cyclin D and p21, which act as downstream effectors of various growth promoting signals. That such factors are in fact control parameters for a bifurcation has been implicated in these studies and is by no means original to our work; see, for example, Figure 7 in PMID: 24904735, where the authors illustrate how growth signals control the G1/S transition by modulating Cyclin D, which acts as a control parameter for the bifurcation. Overall, we believe that this well-rehearsed assumption is at the very least highly plausible and broadly accepted in the field.

The primary innovation of our work lies in the *dynamics* of G1 lengthening. Prevailing models often assume that the control parameter changes in a time-dependent manner to drive cell cycle. In contrast, we propose a "critical lengthening" mechanism, where the control parameter is held in a static but fluctuating state, which itself depends on various growth signals. This alternative mechanism makes a distinct and testable prediction: As the system approaches the critical point, the G1 duration should lengthen in a sharp, exponential manner, accompanied by a dramatic amplification of cell-to-cell variability.

From this perspective, the "open-loop" perturbation experiments in hESCs by Jang et al. (where researchers externally controlled the level of Wnt signaling) are not a limitation but rather an ideal test case. By systematically titrating Wnt signaling, their experiments probe the system's behavior both far from and near the critical point, which would be difficult to observe in a system under a "closed-loop" homeostatic control. While we agree that generalizing from a single experimental system requires caution, the data from hESCs are in good agreement with the unique predictions of our proposed critical lengthening mechanism.

Indeed, in our revised manuscript, we have substantially bolstered the biological grounding of our model by connecting it directly to recent experimental findings. A key addition is a new section and figure (Fig. 2) that explores a core prediction of our theory: the emergence of a prolonged, fluctuating, and reversible "intermediate state" in cells with long G1 phases. We show that this behavior, which within our framework is a consequence of critical slowing down near the G1/S bifurcation, provides a robust mechanistic explanation for the intermediate Rb-E2F activity state recently reported by Konagaya et al. (Nature, 2024) in slowly-dividing cells under growth-inhibiting conditions. Our model not only explains qualitatively this phenomenon but also quantitatively reproduces key features, including the state's reversibility upon perturbation – a feature difficult to explain based on current modeling schemes. These comparisons are summarized in the following discussion in the revised text, as well as a new figure:

A key prediction of our model is that, when poised in the vicinity of the critical point ($C \approx C_{\text{crit}}$), the system can enter a prolonged and fluctuating intermediate state before committing to S-phase. This behaviour is due to critical slowing down of the dynamics (Fig. 2A), a phenomenon also known as a "ghost" transient (Karin, Miska, et al. 2023; Koch, Nandan, Ramesan, and Koseska 2024; Strogatz 2024). These transients have a distinct phenomenology: G1 progression becomes multi-phasic, with a long and highly variable initial phase at an intermediate Rb-E2F activity level, followed by a rapid and more uniform transition into S-phase. Our model predicts that the duration of this intermediate phase is highly sensitive to the precise level of the mitogenic signal, C , and that this state is reversible; cells lingering here have not yet passed an irreversible commitment point and will revert to quiescence upon receiving inhibitory signals.

This predicted phenomenology is in excellent agreement with recent experimental findings on the Rb-E2F activity state in slowly dividing cells (Konagaya et al. 2024). Our model recapitulates the key features observed experimentally, including a prolonged and noisy plateau at an intermediate activity level (Fig. 2B, compare with Fig. 2C in (Konagaya et al. 2024)) and a uniform exit from this state to the S transition (Fig. 2C, compare with Fig. 2E in (Konagaya et al. 2024)). Moreover, our simulation of an inhibitory perturbation correctly predicts that cells in the intermediate state can either revert to quiescence or proceed to S-phase, depending on their position along the trajectory at the moment of inhibition (Fig. 2D, compare with Fig. 2K in (Konagaya et al. 2024)). Collectively, these results suggest that ghost state dynamics provide a robust mechanistic explanation for the experimentally observed intermediate state that safeguards proliferation commitment.

The new figure 2:

Figure 2

Figure 2. Model simulation of the intermediate Rb-E2F state and its reversibility.

(A) Bifurcation diagram of the model. The purple arrows illustrate the trajectory of a cell near the critical point, where it experiences a significant "ghost delay" as it passes the remnant of the stable and unstable fixed points. We mark on the diagram three thresholds (dashed gray lines) that characterise the dynamics - an active activity threshold marking the transition entering the ghost region (corr. to an active state of the Rb/E2F network in (Konagaya et al. 2024)), a high Rb/E2F activity threshold marking the exit from the ghost region (corr. to CDK2 activity of 0.65 in (Konagaya et al. 2024)) and an activity threshold associated with S phase entry. (B) Average trajectories of the Rb-E2F switch activity (proxied by $1 - \tanh(r)$) for cells binned by their total G1 length (coloured lines), aligned to the moment of S-phase entry ($t=0$). Lighter, thin lines show a single sample trajectory from each bin to illustrate cell-to-cell variability. (C) Histograms showing the distribution of time spent in two qualitatively distinct phases of G1 for a population of simulated cells. The top panel shows the distribution of durations from the initial crossing of the active threshold to the initial crossing of the high activity threshold. The bottom panel shows the subsequent, more rapid transition from the high activity threshold to S phase threshold. (D) Simulation of an inhibitory perturbation. Cells progressing through G1 are subjected to a sudden drop in C at $t=15$ h (dashed line). Trajectories show that cells that have not yet passed a commitment point revert to a quiescent state (red), while committed cells proceed to S phase (black), demonstrating the reversibility of the active-intermediate state. See the Appendix for simulation details and parameters.

For completeness, we have included here Fig. 2 from Kongaya et al., which contains the experimental data to which our narrative above and in the revision relates:

a, Single-cell traces of CDK2 and E2F activities. Circles, CDK2-activated or E2F-activated timing. **b**, Cumulative frequency of CDK2-activated, E2F-activated and S phase-entered cells. $n = 5,969$ cells; 1 out of 3 biological replicates. **c**, Cell traces were computationally aligned at S phase entry and stratified based on the variable time cells spend from E2F-active to S phase entry. CDK2 and E2F activity traces (mean per cell population). $n = 244, 246, 226$ and 125 cells for 5–10 h, 10–15 h, 15–20 h and 20–25 h, respectively; 1 out of 3 biological replicates. For **a–c**, conditions were release with growth medium + CDK4/6i ($1 \mu\text{M}$). **d**, Data in **c** plotted as a phase-plane trajectory. **e**, Data in **c** analysed for the variable time cells spend from CDK2-active to CDK2 activity = 0.65 (top), and from CDK2 activity = 0.65 to S phase entry (bottom). Dashed lines indicate the median. **f**, Left, percentage of cells with E2F activation by 40 h after release with starvation medium + EGF (20 or 0.2 ng ml^{-1}) \pm CDK4/6i ($1 \mu\text{M}$). Right, percentage of S enter, E2F reverse and undecided cells among E2F-activated cells (mean \pm s.e.). Cells were categorized based on behaviours until 40 h after release (see Methods for more detail). $n = 2,655, 3,611, 1,685$ and $3,042$ cells for EGF 20 , EGF 0.2 , EGF $20 + 4/6i$ and EGF $0.2 + 4/6i$, respectively; 3 biological replicates. **g**, Single-cell traces of E2F activity in cells categorized in **f**. **h**, Single-cell traces of E2F activity in S enter and E2F reverse RPE-1 cells (left), determined based on DNA content versus 5-ethynyl-2'-deoxyuridine (Edu) incorporation at the end of live-cell imaging (right). One out of 3 biological replicates. **i**, CDK2 and E2F activity traces (mean \pm s.e.) after release with starvation medium + EGF (20 ng ml^{-1}) + CDK4/6i ($1 \mu\text{M}$). $n = 534$ (DMSO) and 476 (EGFR inhibitor (EGFRi)) cells; 1 out of 3 biological replicates. **j**, Data in **i** plotted as a phase-plane trajectory. **k**, Single-cell traces of CDK2 activity before and after EGFRi. G1 and S/G2 cells in **i** were gated based on the CRL4^{Cdt2} reporter signal. $n = 5$ cells each; 1 out of 3 biological replicates.

Taken together we believe that these findings indicate that our proposed framework constitutes, at the very least, a plausible mechanism underlying the observed G1 lengthening processes.

Regarding the second concern, we believe that embedding this control mechanism within a negative feedback loop is also highly plausible. For example, during terminal differentiation, in many systems the negative feedback component serves to arrest the cell cycle by extending G1. Unfortunately, for most systems, we simply do not know the precise mechanisms underlying stem cell homeostasis and their control by signaling. Indeed, a principal aim of the current theoretical scheme is to provide a unifying framework that can guide experimental investigation into these programs through its predictions.

Actually, the observation that dividing cells typically control their own numbers in vivo through negative feedback on renewal is itself rather speculative, whereas the alternative that dividing

cells are controlled by their differentiated offspring has both experimental and theoretical support (which the authors, incidentally, fail to cite).

We thank the reviewer for this important point, and we appreciate the suggestion to consider this alternative. While our main text focused for clarity and simplicity on a minimal single-compartment model, the core principles of our theory are not contingent on the specific feedback structure – a point that we should have emphasized more clearly in the original manuscript. To demonstrate this explicitly, we have now added a new section to the Methods ("Generality of the Feedback Mechanism: Control by Differentiated Cells") – see below. There, we analyze a two-compartment model, where the feedback signal is generated by the differentiated cell population, as proposed by the reviewer. Our analysis shows that this alternative regulatory scheme leads to precisely the same qualitative conclusions. A homeostatic steady-state still requires that the net growth rate of the stem cell pool be zero ($dN/dt=0$), which in turn necessitates that the system operates at an average G1 length T_s where the bias in self-renewal vs. differentiation is balanced ($\theta(T_s)=0$). As any such prolonged G1 duration can only be achieved near the bifurcation, and the system is therefore once again forced to self-tune to the critical point. The stability analysis remains identical, yielding the same RD and DR topologies based on the sign of the feedback (while noting that in general the introduction of delays in feedback may lead to oscillations around the fixed point – see also the reply to referee 2). We have also incorporated what we believe are relevant citations to the literature on this topic, as requested by the reviewer. If the reviewer feels that we are missing further important citations, we would of course welcome their suggestions.

4 Generality of the Feedback Mechanism: Control by Differentiated Cells

In the main text, we considered a feedback loop where the undifferentiated stem cell population, N , directly regulates its own size. However, it is also plausible that feedback is mediated by the differentiated cell population, D . Here, we show that our model's conclusions are robust to this alternative regulatory structure.

Let us consider a two-compartment model. Stem cells, N , divide with a G1 duration T . Each division produces, on average, $(1 - \theta(T))$ differentiated cells and $(1 + \theta(T))$ stem cells. The differentiated cells, D , are removed (e.g., die) at a fixed rate, γ . The system dynamics can be described as:

$$\frac{dN}{dt} = \frac{N}{T} \theta(T) \quad (9)$$

$$\frac{dD}{dt} = \frac{N}{T} (1 - \theta(T)) - \gamma D \quad (10)$$

Now, let's assume that the differentiated cells, D , secrete a ligand, l , at a rate p_D , which is then degraded. At steady-state, the ligand concentration is proportional to the size of the differentiated population, $l_{ss} \propto D$. If this ligand controls the stem cell cycle parameter C , then the feedback function becomes $C = f(D)$.

For the system to reach a homeostatic steady-state, both populations must be constant ($\dot{N} = 0$ and $\dot{D} = 0$). The condition $\dot{N} = 0$ requires that $\theta(T_s) = 0$ at some steady-state G1 length T_s . The condition $\dot{D} = 0$ then implies that, at this steady-state, $N_s/T_s = \gamma D_s$. This shows a stable coupling between the two population sizes. As in the single-compartment model, any homeostatic state that relies on a prolonged G1 duration, T_s , can only be achieved if the system is tuned to the vicinity of the critical point, where G1 length is exquisitely sensitive to the control parameter.

Crucially, the stability analysis and the emergence of the RD and DR topologies remain unchanged. The feedback loop now passes through the differentiated compartment, but the logic is identical: a perturbation in N affects D , which in turn affects C and feeds back on N . For example, if the ligand is a mitogen ($C \propto D$), an increase in N leads to an increase in D , which increases C and shortens T . This corresponds to $df/dN > 0$, requiring the DR topology for stability.

Therefore, the core mechanism of self-tuning to criticality and the resultant trade-offs between rapid growth and mutant rejection are general principles that hold regardless of whether the feedback is mediated directly by stem cells or indirectly by their differentiated progeny.

Finally, while we agree that the mechanism of negative feedback mediated by differentiated cells is likely to be widespread, we would respectfully disagree that the existence of mechanisms by which cells control their own numbers by negative feedback is speculative (see for example PMID:25171404, PMID: 30581080). In practice, it is likely that feedback loops involve many pathways. However, the only important constraint for our model is that the defined steady-state of the self-renewing population occurs when the duration of the cell cycle is prolonged.

Finally, the idea that mutant cells fail to overtake normal populations because mutant cells are made to divide more slowly lacks any experimental support that this reviewer is aware of. What little we do know about how mutant cells are prevented from overtaking tissues suggests that active processes involving cell recognition and cell killing are most often involved. In the end we are left with a clever theory, but not a lot of motivation for accepting it.

We are grateful to the reviewer for raising this point, which allows us to clarify a central aspect of our model. We agree with the reviewer that a model proposing that all mutant cells are simply made to divide more slowly would lack experimental support. However, our framework does not make this assumption. Instead, our model explores the consequences of mutations that can either accelerate or decelerate a cell's passage through G1. The outcome of such mutations is entirely dependent on the underlying circuit topology.

In the RD topology, slower-dividing cells are at a self-renewal disadvantage because extended G1 durations increase their likelihood of differentiation. This is biologically sound, as terminal differentiation in many contexts exhibits this property (see, e.g., PMIDs: 7863328, 32553172).

Our proposal for robust, long-term homeostasis lies in the DR topology. Here, the logic is inverted: a shorter G1 phase is coupled with a higher probability of differentiation, while a longer G1 is coupled with self-renewal. This creates a passive mechanism for mutant cell rejection, where fast-cycling mutant cells are intrinsically disadvantaged because they are preferentially removed through differentiation. One example, described in the manuscript, is HSCs. This system displays the key hallmarks of the DR topology. It is known that forcing HSCs into cell cycle can lead to their depletion, consistent with rapid division being coupled to a loss of self-renewal potential. In addition, during aging, the hematopoietic system shows a slow accumulation of clonal mutants that are characterized by impaired sensitivity to mitogens and longer cell cycle times, which is precisely the pattern of mutant selection predicted by the DR mechanism.

Another plausible instantiation of the DR mechanism is provided in the revised discussion, where we demonstrate how established features of spermatogenic stem cell self-renewal hint at a DR-based mechanism for mutant cell elimination:

Our model reveals a fundamental trade-off between the RD and DR topologies. The RD topology, while enabling rapid population recovery from low cell numbers, is inherently fragile to invasion by "cheater" mutants with shorter G1 phases. Conversely, the DR topology is robust against such mutations, as it preferentially removes faster-cycling cells through differentiation before they can self-renew. However, this robustness comes at a significant cost: a severely limited ability to recover from population bottlenecks.

These inherent fragilities may be overcome by complementary mechanisms. For the RD topology, for instance, biphasic regulation, where growth becomes negative at very fast division rates, could prevent mutant accumulation, albeit at the risk of population collapse (Karin and Alon 2017). For the DR topology, recovery from low population abundances could be boosted by periodic signals that force cells into the cycle. Such a mechanism is observed in spermatogenesis, where the seminiferous cycle provides a periodic signal that drives stem cell division approximately every 10 days. Following division, these stem cells appear to enter a transient state that is more susceptible to differentiation but can be rescued by self-renewal factors present at that stage of the cycle (Nakagawa et al. 2021). This process may potentially act as a variation of the DR topology: it prevents runaway "cheater" mutants that escape the seminiferous cycle's control, while the periodic nature of the signal ensures the population can recover from large perturbations, as has been observed experimentally (Kitadate et al. 2019).

Finally, our proposed mechanism offers a framework for passive mutant cell elimination that complements, rather than replaces, active processes like cell-cell killing, and makes specific, testable predictions.

In addition, several components of the paper that are presented as new theoretical results, while convincing, are either not that relevant or not that much of an advance over previous work. For example, the lengthy discussion about Rb and cyclins in the first half of the paper is interesting, but not particularly relevant to the rest of the paper (i.e. the details of the circuit don't particularly matter to the remainder of the paper, other than it is characterized by bistability).

We agree that the core novelty of our paper is not the existence of the Rb/E2F bistable switch, which is indeed a well-established concept. Rather, the purpose of our detailed exposition of this regulatory circuit is to firmly ground our subsequent theoretical analysis in a concrete and well-characterized biological context. We believe that this is essential for two reasons:

1. Establishing plausibility: By showing how our abstract model parameter, C , directly corresponds to key molecular players like Cyclin D, we establish that the control mechanism we propose is not arbitrary but is based on the known targets of mitogenic signaling pathways.
2. Enabling direct comparison with experimental data: Using this specific circuit allows us to move beyond abstract theory and make direct, quantitative comparisons with experimental findings. A key example is in our new Figure 2. Here, we show that the model's predicted "ghost transient" dynamics (a direct consequence of its structure) faithfully recapitulates signature features of the intermediate Rb-E2F activity state recently observed in slowly-dividing cells (Konagaya et al., 2024). This validation would be impossible without the specific background on the underlying circuit.

Our primary theoretical advance in the first part of the manuscript lies in proposing a novel dynamic mechanism for G1 lengthening that is based on fixing the control parameter and allowing for fluctuation-driven cell cycle entry. Therefore, the initial discussion of the Rb/E2F circuit serves the role of connecting our novel theoretical insights to tangible, measurable biology.

In addition, the general idea that "self-organized criticality" should characterize biological systems became very popular in physics twenty years ago but hasn't achieved much traction in the intervening time. It's not clear that this work adds a tremendous amount more to that discussion.

We thank the reviewer for this perspective and wish to clarify a key distinction. Our work does not start with the premise of "self-organized criticality" as a generic and abstract paradigm but relates to a specific and motivated mechanism of self-tuning to a critical point driven by negative feedback and resource competition. As our work is driven by experimental observations, provides new interpretations of existing experimental phenomenology, and makes specific testable predictions, we believe it stands as a tangible and productive application of critical dynamics to biology, offering specific hypotheses that to our knowledge move beyond the more abstract discussions and literature to which the reviewer refers.

Finally, it is a shame that the manuscript is written in a manner that makes it relatively inaccessible to most biologists. Phenomena that are described in the manuscript tend to be derived from first principles analyses of equations, even when the phenomena have other examples in biology that could be used to illustrate them. For example, critical slowing when systems cross bifurcations is a generic phenomenon, and appears in other areas of biology, such as ecosystem science, where helpful analogies could have been drawn. Greater use of analogies and simulations, such as those that appear in Fig. 2C and D, could have also made the early results of the manuscript more accessible, and could have spared biologists from having to wade through stochastic and partial differential equations that, in the end, are only important in their broadest interpretation.

We thank the reviewer for this constructive and important feedback. We agree that making theoretical concepts accessible to a broad biological audience is essential, and we have made significant revisions to the manuscript with this goal in mind.

A major addition to the manuscript, directly inspired by the reviewer's suggestion, is a new section and figure (Figure 2) dedicated to visualizing the dynamics of "critical slowing down". Instead of relying on a purely mathematical description, we now use simulations to illustrate this phenomenon as a "ghost transient"—a prolonged, reversible intermediate state. Crucially, we directly compare results of model simulations to recent experimental findings on the Rb-E2F switch (Konagaya et al., 2024), providing a concrete and timely biological example that we believe makes the abstract concept more tangible.

Furthermore, to better illustrate how our model can be realized biologically, we have added a new section to the Appendix ("*Mechanistic Examples of Population Feedback*"). This provides concrete examples of simple biological circuits, such as competition for a diffusible ligand, that would naturally give rise in the distinct RD and DR topologies we describe. Finally, we have refined our narrative throughout, especially in the context of the modelling, to improve its clarity and accessibility.

Ultimately, the combination of being highly speculative while also being difficult for experimentalists to wade through is likely to diminish the potential impact of this work, which does contain some interesting ideas that are potentially worth sharing.

We thank the reviewer for their input and believe that the revised manuscript has addressed their concerns.

Reviewer #4:

In this manuscript, the authors develop a theoretical framework to analyze how coupling between cell cycle progression (specifically G1 lengthening) and cell fate decisions enables robust tissue population control by stem cells. By modeling the G1-S transition as a noisy saddle-node bifurcation-where mitogens acting on cell cycle proteins, such as cyclin D and cyclin E, act as control parameters for the G1-S transition-the authors demonstrate that competitive feedback self-tunes the system to a critical bifurcation point where dynamics slow down. This critical positioning confers two key advantages: (1) robust population (of stem cells) set-point maintenance and (2) efficient rejection of mis-sensing mutants. The authors test their model predictions using experimental data from human embryonic stem cells (hESCs) under varying Wnt signaling conditions including non-linear G1 lengthening and amplified variability near the critical point.

Overall, the authors bring to light two interesting and somewhat counterintuitive predictions: (1) positioning of stem cells near the critical point (where cell-cell variability is amplified) for G1-S transition can confer robust population control, and (2) there might be an inherent trade-off between the ability of stem cells to rapidly regenerate a tissue and avoiding mutant takeover. However, there are a few major concerns that need to be addressed to significantly strengthen the manuscript as highlighted below.

We thank the referee for their positive endorsement of our work and for their constructive recommendations, which we address below.

Major concerns:

1. Robustness of results to choice of parameter values is unclear: Manuscript code and detailed tabulation of parameter values used for each figure are missing. For example, parameter values for V , reasoning for a choice of 2 for the Hill coefficient for the cooperative Rb phosphorylation via the E2F/CycE cascade ($\gamma P \propto 1/(1+(r/k_2)^2)$), and initial conditions for simulations are not mentioned. This is critical for both reproducing the authors' results and for testing how the choice of parameter values impacts the results. For this purpose, the authors should tabulate the parameter values used for each figure (maybe in the figure captions or the supplementary section) and comment about the range of parameter values for which their results hold.

To address all the points raised, we have added a new, detailed appendix section titled "Appendix: Simulation Methodology and Parameters."

This new section provides:

1. A step-by-step description of the agent-based simulation methodology.
2. A comprehensive table of all parameters used in our simulations (including V , initial conditions like r_0 , and values for each figure), making the work fully reproducible.

3. A dedicated subsection justifying our choice of a Hill coefficient of 2 for the Rb phosphorylation term, explaining its role in establishing the necessary bistability for the model.

Regarding the robustness of our results, while the specific quantitative outcomes will vary with parameter choices, the core qualitative conclusions of our paper are robust. This is because our findings depend not on the particular parameter values, but on the universal dynamical properties of a system operating near a saddle-node bifurcation. As we now clarify in the appendix, any model that exhibits this fundamental feature will display the behaviors of critical slowing down and the trade-offs we describe. We also made the simulation code available (<https://github.com/karin-lab/cellcycle>).

2. Non-linear G1 lengthening in hESCs upon Wnt removal is not sufficiently supported: In Fig. 1D, the sharp non-linear increase in G1 length of hESCs when Wnt activity/mitogen signaling is reduced needs to be better supported, especially because the accuracy of fitting for the authors' model is not quantified. One line of evidence to better support this claim would be to (1) quantify the strength of fitting for the authors' model (say R^2 value), and (2) perform a linear fit and compare the R^2 values for the linear model to the authors' model.

We are grateful for this suggestion. To better support our claim of critical G1 lengthening, we have performed the quantitative analysis recommended by the reviewer and have presented it in the main text and in a new Supplementary Figure:

In our model, activation of the Wnt pathway modulates the control parameter C (Figure 1D). As mitogenic signalling approaches a critical threshold, the model predicts a nonlinear rise in mean G1 duration, accompanied by a sharp increase in cell-to-cell variability. Experimental results from Jang *et al.* on hESCs under varying Wnt conditions confirm these predictions: both the mean G1 length and its variance diverge within a narrow range of Wnt activity, in line with model predictions (Figure 1E). The bifurcation-based model captures the experimental data for mean G1 length ($R^2 = 0.86$) and variance ($R^2 = 0.92$), substantially outperforming linear regressions ($R^2 = 0.62$ and $R^2 = 0.34$, respectively; Figure EV1).

Figure 6

Figure EV1: Quantitative Comparison of G1 Lengthening Models

Comparison of experimental data from Jang et al. (Jang et al. 2019) with model predictions for the mean (left) and variance (right) of G1 length in hESCs under different Wnt signalling conditions. The model, based on a noisy saddle-node bifurcation (solid black line), provides a strong fit for both the mean ($R^2 = 0.86$) and the variance ($R^2 = 0.92$), outperforming linear regressions (dotted gray line, $R^2 = 0.62$ and $R^2 = 0.34$, respectively). Experimental data points show bootstrapped means with 95% confidence intervals. The dashed vertical line indicates the bifurcation point.

3. Self/fine-tuning of the system near the critical point and corresponding population control for RD and DR topologies should be characterized for smaller stem cell niches: The authors highlight that cell-cell variability is amplified near the critical point (Fig. 1E) and fine-tuning near the critical point is necessary for population control (as finite T_s values occur close to C_{cric}). It is somewhat counterintuitive that robust population control can only be achieved at the same point where cell-cell variability is also enhanced. A natural question is what happens if the stem cell niche is much smaller (say ~ 10 cells)? Would the increased cell-cell variability near the critical point prevent population control for smaller stem cell niches?

We addressed this important point by performing new simulations for smaller stem cell niches of approximately 10 cells. These simulations are now given in a new supplementary figure. As shown in the figure, our results confirm that while smaller niches exhibit larger population fluctuations due to increased stochasticity, the fundamental mechanism of population control is preserved. The system consistently tunes its behavior towards the critical point, demonstrating that the negative feedback loop is robust enough to maintain stability even in these conditions.

Figure 7

Figure EV2: Robust Population Control in Small Stem Cell Niches

To test the robustness of our proposed control mechanism, we simulated the population dynamics for both the RD (left) and DR (right) topologies in a small stem cell niche with a target set-point of $N = 10$ cells. The simulations show that even in the presence of significant demographic fluctuations inherent to small populations, the negative feedback loop effectively maintains the population size around the set-point. In both topologies, the system consistently self-tunes to the vicinity of the critical point (bottom right panels), demonstrating that the control mechanism is robust and does not require a large population size to function.

Minor concerns:

1. Fig. 1C is unclear -> the "delay" black line should be explained in more detail to contextualize the stretched exponential fit that predicts G1 lengthening in the vicinity of C_{crit}

We realize that the figure was indeed confusing. We now clarify that the black line corresponds to the exact analytical solution for G1 based on the delay of the saddle node bifurcation model.

2. Legend/colors in the population size plot in Fig. 3C seems to be flipped

We thank the referee for pointing this out and have now fixed this in the manuscript.

3. It should be made explicit in the Introduction that the paper focuses on stem cell self-renewal vs differentiation as opposed to population control of different types of differentiated cells. In line with this, first sentence of the abstract could be changed to "tissue homeostasis requires a precise balance between stem cell self-renewal and differentiation".

We agree and have fixed this in the introduction and abstract.

25th Sep 2025

Manuscript Number: MSB-2025-13167R

Title: Cell cycle criticality as a mechanism for robust cell population control

Author: Benjamin Simons

Omer Karin

Dear Dr Karin,

Thank you for sending us your revised manuscript. We have now heard back from the three reviewers who agreed to evaluate your revised study. As you will see below, the reviewers are satisfied with the performed revisions and support publication. Before we can proceed with formal acceptance, we kindly ask you to address the following remaining issue:

1. Regarding the remaining comment raised by Reviewer #1, we recommend that the relevant code be deposited in a public repository.

On a more editorial level:

2. Please provide up to five keywords in the manuscript file.

3. Remove the "Authors' Contributions" section from the manuscript.

4. Remove the "Declaration of Interests" section.

5. The "Methods" section and the two associated references should be included directly in the manuscript file. The references must follow the Molecular Systems Biology reference style. Please remove the "Appendix" file, along with all corresponding callouts in the manuscript text.

6. Source Data: Please remove the sentence "Source data is available at Ref. (Jang et al. 2019)" from the legend of Figure 1. There is no need to provide source data for these two figure panels.

7. Please add the missing callouts for Figure 5A-B in the manuscript text.

8. Please ensure that the manuscript sections are properly titled and ordered as follows:

Title Page - Abstract - Keywords - Introduction - Results - Discussion - Methods - Data Availability - Acknowledgements - Disclosure and Competing Interests Statement - References - Figure Legends - Table(s) - Expanded View Figure Legends.

9. Please address the following issues related to figure legends:

- Please note that the box plots need to be defined in terms of minima, maxima, centre, bounds of box and whiskers, and percentile in the legend of figure 1D

- Please note that information related to n is missing in the legends of figures 1D, EV1

- Please note that the measure of center for the error bars needs to be defined in the legend of figure EV1

Click on the link below to submit your revised paper.

Sincerely,
Jingyi

Jingyi Hou, PhD
Senior Editor
Molecular Systems Biology

If you do choose to resubmit, please click on the link below to submit the revision online before 25th Oct 2025.

*** PLEASE NOTE *** As part of the EMBO Press transparent editorial process initiative (see our Editorial at <https://dx.doi.org/10.1038/msb.2010.72> , Molecular Systems Biology will publish online a Review Process File to accompany accepted manuscripts. When preparing your letter of response, please be aware that in the event of acceptance, your cover letter/point-by-point document will be included as part of this File, which will be available to the scientific community. More information about this initiative is available in our Instructions to Authors. If you have any questions about this initiative, please contact the editorial office (msb@embo.org).

Reviewer #1:

The original manuscript was thoroughly reviewed by four referees, and the authors have carefully and fully responded to all the criticisms and suggestions of the referees. The paper is much improved by all the revisions. As everyone agrees, the work is very theoretical and 'conceptual'. In my opinion, the work is sufficiently grounded in molecular cell biology to warrant publication in MSB. The dynamical systems approach adopted by the authors is a well-respected and influential tool of systems biologists. My only further suggestion is that the authors include in the Supplementary Information some representative code (in Matlab or another common language) for implementing their models. It's easy to do and a great help to anyone who might want to pursue their ideas.

Otherwise the paper is ready for publication in MSB.

Reviewer #2:

The authors have addressed all of my previous comments and I recommend publication.

Reviewer #4:

The authors have satisfactorily addressed this reviewer's concerns.

Dear Dr. Hou,

Thank you for the positive evaluation of our manuscript. We have addressed the final remaining issues as detailed below.

1. Code Availability: The full analysis and simulation code is deposited in a public GitHub repository, accessible at: <https://github.com/karin-lab/cellcycle>. We have clarified its availability in the revised Methods section.
2. Keywords: We have added five keywords to the manuscript as requested.
3. Authors' Contributions: This section has been removed.
4. Declaration of Interests: This section has been removed.
5. Methods Section: The Methods section and its references have been moved from the Appendix into the main manuscript file. The Appendix file and all corresponding callouts have been removed.
6. Source Data in Figure 1: The sentence regarding source data availability has been removed from the legend of Figure 1.
7. Figure 5 Callouts: We have added the missing callouts for Figure 5A-B in the main text.
8. Manuscript Order: We have re-ordered the manuscript sections to match the journal's required format.
9. Figure Legends: We have revised the figure legends as follows:
 - The legend for Figure 1D now defines the components of the box plot (median, quartiles, whiskers, and fliers).
 - Sample sizes (n) have been added to the legends for Figure 1D and Figure EV1.
 - The legend for Figure EV1 now clarifies that the center of the error bars represents the bootstrapped mean.

Best,

Omer Karin

1st Oct 2025

Manuscript number: MSB-2025-13167RR

Title: Cell cycle criticality as a mechanism for robust cell population control

Dear Dr Karin,

Thank you again for sending us your revised manuscript. We are now satisfied with the modifications made and I am pleased to inform you that your paper has been accepted for publication.

Sincerely,
Jingyi

Jingyi Hou, PhD
Senior Editor
Molecular Systems Biology
